# Difficulty Orientations, Gender, and Race/Ethnicity: An Intersectional Analysis of Pathways to STEM Degrees

**Samantha Nix [1] and Lara Perez-Felkner [2],\*** 

[1]    Academic Center for Excellence, Florida State University, Tallahassee, FL 32306, USA; snix@fsu.edu
[2]    Department of Leadership and Policy Studies, College of Education, Florida State University, Tallahassee, FL 32306, USA
\*    Correspondence: lperezfelkner@fsu.edu

**Abstract:** Is there a relationship between mathematics ability beliefs and STEM degrees? Fields such as physics, engineering, mathematics, and computer science (PEMC) are thought to require talent or brilliance. However, the potential effects of difficulty perceptions on students' participation in STEM have yet to be examined using a gender and race/ethnicity intersectional lens. Using nationally representative U.S. longitudinal data, we measure gender and racial/ethnic variation in secondary students' orientation towards mathematics difficulty. We observed nuanced relationships between mathematics difficulty orientation, gender, race/ethnicity, and PEMC major and degree outcomes. In secondary school, the gap between boys' and girls' mathematics difficulty orientations were wider than gaps between White and non-White students. Mathematics difficulty orientation was positively associated with both declaring majors and earning degrees in PEMC. This relationship varied more strongly based on gender than race/ethnicity. Notably, Black women show higher gains in predicted probability to declare a mathematics-intensive major as compared to all other women, given their mathematics difficulty orientations. This study's findings show that both gender and racial/ethnic identities may influence the relationship between mathematics difficulty orientation and postsecondary STEM outcomes.

**Keywords:** ability beliefs; higher education; gender; race and ethnicity; STEM degrees; intersectionality; STEM education; URM students

## 1. Introduction

Science, technology, engineering, and mathematics (STEM) fields have been associated with intellectual giftedness or brilliance, particularly in mathematics-intensive science fields (Snow 1961; Lubinski et al. 2001; McPherson 2017; Meyer et al. 2015). Women and racial/ethnic minorities have been shown to be as strong and able in mathematics as their male and majority-group peers, countering biologically-driven hypotheses about their inferiority (e.g., AAPA 1996; Ceci et al. 2009, Hyde and Linn 2006). Still, as suggested by research on implicit biases (Nosek et al. 2009), even false perceptions have real-world consequences (Merton 1995). In the most mathematics-intensive STEM fields, significant participation gaps remain (Schneider et al. 2015). Women and racial/ethnic minority students' performance on mathematics tasks may be negatively affected by stereotype threats that associate mathematics ability with groups other than their own (Steele 1997; Beilock 2008). Because women and racial/ethnic minority groups are the least represented in mathematically-intensive STEM fields (NSF 2015; Anderson and Kim 2006; Corbett and Hill 2015), it is important to understand how students' beliefs have contributed to gender and racial/ethnic disparities in postsecondary degrees in these fields.

There is a high need for graduates with computational skills (BLS 2015; Xue and Larson 2015). Moreover, STEM career fields have more stable employment rates and higher initial pay for women and underrepresented racial/ethnic groups (Langdon et al. 2011; Olitsky 2014). Broadening participation in science and technology fields therefore bears relevance for economic competitiveness (Augustine 2005) as well as equity, especially for women and underrepresented racial/ethnic minority groups who remain economically disadvantaged compared to White men (DeNavas-Walt and Proctor 2014). For girls, and perhaps especially for girls from underrepresented backgrounds, stereotype-related challenges may especially present themselves when working with mathematics perceived as difficult (Dweck 2007).

This study investigates the long-term consequences of high school students' beliefs about their mathematics ability with difficult tasks and materials. More specifically, we measure the relationships between *difficulty orientation* and degree field in mathematics-intensive postsecondary majors. Here, we use *difficulty orientation*, to describe students' perceptions of their ability with difficult material specifically, rather than their general perceived ability. Using an intersectional lens throughout this manuscript (Crenshaw 1989; Crenshaw 1991), we examine how this relationship varies by gender and race/ethnicity. Using nationally representative Education Longitudinal Study 2002/12 (ELS) data including high school and college transcript records, this study poses the following questions: (1) do difficulty orientations differ by gender and race/ethnicity, (2) to what extent do difficulty orientation measures predict mathematics-intensive STEM degrees, and (3) do the relationships between difficulty orientations and degree field differ by gender and race/ethnicity?

## 1.1. Mathematics-Intensive STEM Fields, Gender, and Race/Ethnicity

Mathematics-intensive science fields have been a recent research focus given their capacity to develop graduates with needed computational skills and their low representation of women and people from underrepresented minority groups (Ceci et al. 2009; Corbett and Hill 2015; Perez-Felkner et al. 2012). In 2012, women represented only about 20% of U.S. bachelor's degree holders in engineering, computer science, and physics combined (NSF 2015). Black, Latino, and Native American populations represented 11% of mathematics and statistics graduates, 12% of physical sciences graduates, 13% of engineering graduates, and 20% of computer science graduates (NSF 2015). This study continues the research on mathematics-intensive fields through an examination of postsecondary degree outcomes in the physical sciences, engineering, mathematics, and computer science fields (PEMC) (Perez-Felkner et al. 2012). This study builds on this emerging line of research, with particular attention to the intersections between race/ethnicity and gender.

Intersectionality as a field of study invites researchers to actively study the overlap of identity categories, and the ways that these categories may indicate disadvantages to specific groups (Cho et al. 2013; Crenshaw 1989, 1991). Much STEM research has been conducted using this approach. For instance, some have chosen to look at intersectional identities with gender (Harper et al. 2011; Johnson 2011; Strayhorn 2015; Charleston et al. 2014), while others have examined both gender and race/ethnicity at the intersection of some other construct, such as STEM attitudes (Else-Quest et al. 2013), engineering confidence (Litzler et al. 2014), STEM stereotypes (O'Brien et al. 2015), and engineering learning outcomes (Ro and Loya 2015). Yet, within engineering research specifically, scholars criticize the lack of intersectionality research that uses even the most basic methodological techniques such as analyzing both gender and race/ethnicity within one study (Beddos and Borrego 2011). This research builds on previous work by looking at gender and race/ethnicity separately before integrating a gender and race/ethnicity intersectional approach through interaction terms.

First, we turn to student and school characteristics prior to college. While women continue to be underrepresented in science degrees irrespective of their comparatively high achievements in secondary school (Nord et al. 2011), analyses of nationally representative longitudinal cohort data from ELS suggest racial/ethnic disparities in STEM higher education appear primarily a function of academic preparation (Perez-Felkner et al. 2014). In a related study with ELS data, when holding academic preparation constant, Black women were twice as likely as White women to declare

physical science/engineering majors, and Black and Latino men were more likely than White men (Riegle-Crumb and King 2010).

Advanced high school mathematics and science courses are associated with majoring in undergraduate STEM fields (Reilly et al. 2015; Hanson 2004), including engineering (Tyson 2011). Problematically, students' access to these courses varies considerably across and within schools (Fletcher and Tienda 2010; Crosnoe and Schneider 2010). While national enrollments in advanced math and science courses have increased since 1990 across racial/ethnic groups, Asian and White students continue to enroll at higher rates than Latino and Black students (Kena et al. 2014). Black and Latina girls were found to take more advanced high school mathematics course sequences than their male peers (Riegle-Crumb 2006).

Next, we turn to enrollment and experiences within these majors. Decades of research show women tend to underestimate their ability with tasks in stereotypically masculine domains (Beyer 1990; Correll 2001, 2004; Beyer and Bowden 1997). Correspondingly, women are less likely to participate in fields associated with "brilliance" (Meyer et al. 2015; Leslie et al. 2015), including mathematically-intensive STEM fields such as physics. Women of color in STEM may experience a double bind from the combined experiences of being underrepresented by gender and race/ethnicity (Ong et al. 2011). In a study of enrollment and persistence in engineering at nine southeastern public universities, women of all racial/ethnic groups enrolled at proportionally lower rates than expected for their overall representation in college (Lord et al. 2009). Asian women were the closest to meeting proportional representation, while Latina women were the furthest from meeting representational proportions. Consistent with the high school course taking literature, women in this dataset did just as well as men in early STEM courses, and Black women persisted at higher rates than their Black male counterparts (Lord et al. 2009).

Scholars have studied the relationship between postsecondary experiences and STEM degree attainment. Higher admissions selectivity has been negatively associated with Black and Latino STEM persistence (Chang et al. 2014). Large institutional size and research expenditures are also negatively related to women's and racial/ethnic minority students' persistence in STEM fields (Griffith 2010). College student experiences such as faculty–student interaction (e.g., Cole and Espinoza 2008) and engagement (e.g., Brint et al. 2008) have also been associated with positive postsecondary STEM outcomes. National studies have tended not to consider the effects of undergraduate research on sex and gender disparities, even though such opportunities have been found to have positive impacts on student success generally (Kilgo and Pascarella 2015; Thiry et al. 2011; Russell et al. 2007), and would likely be a formative experience for STEM students.

## 1.2. Perceived Ability and Difficulty

Perceptions of ability and difficulty are not new educational concepts; however, we argue that a fresh examination of these ideas is useful to understanding variation in STEM fields. Notably, we discuss *perceptions* of these concepts, rather than objective measures of ability, in the following section and as the variables of main interest in this study. We first describe central theories related to perceived ability: self-concept and self-efficacy. Then, we describe theories and recent studies related to perceived difficulty, particularly through the lens of challenge, talent, and brilliance.

Research on perceived ability and difficulty draws on learning theory models such as self-efficacy and self-concept. Self-concept refers to people's understanding or perceptions of themselves through an evaluation of feedback from others or messages in the environment (Markus and Wurf 1987; Marsh 1990). The connection between self-concept and academic achievement was sharpened about 25 years ago, when Marsh developed the Self-Description Questionnaire and started to describe the *academic* self-concept (Marsh 1986, 1990; Shavelson et al. 1976). Academic self-concept is domain-specific and describes one's perceived ability within a field of study (Marsh 1986; Möller et al. 2009; Möller and Marsh 2013). In contrast, self-efficacy describes perceived ability to do a specific task or fulfill a specific goal. Self-efficacy was described by Bandura (1977) and has been extensively used to frame research on academic achievement and participation in STEM (Pajares 1996; Rittmayer and Beier 2009). Notably,

previous scholars have described the relative difficulty of measuring self-efficacy and the existence of too many similar constructs as limitations for using it in research (Pajares 1996; Zimmerman 2000).

The measures in this study focus on domain-specific and domain-general ability beliefs. Our study particularly emphasizes perceived ability with difficult material. Foundational self-concept research shows that students tend to believe that they have ability in either mathematics or verbal domains, even when objectively successful in both areas (Marsh 1986; Möller et al. 2009). Notably, students appear to see their ability in one domain (such as English) in contrast with another domain (such as mathematics) (Möller and Marsh 2013). Women with high ability in both mathematics and verbal domains appear more likely to enter occupations which reflect their verbal skills (Wang et al. 2013). How domain-specific ability beliefs influence STEM major choice seems a question of merit.

Under the umbrella of these ability beliefs, perceived difficulty bears special importance for women's success in STEM fields. The pervasive cultural model frames these fields as being "hard" (Nix 2018) and requiring innate talent or brilliance (Leslie et al. 2015) to compete for limited opportunities to enter these fields (Corbett and Hill 2015). Masculine-normed language within STEM may reinforce beliefs about the formidable nature of these degree fields (Haswell 2019). The theories and studies described below all focus on concepts related to perceived difficulty. Cultural models have been shown to have real-world consequences (Merton 1995); those countries with greater gender equity ratings but more sex-typed career associations have lower shares of women succeeding in STEM (Sjöberg 2010; Charles 2017; Penner and Cadwallader-Olsker 2012). Beliefs matter. While we use the authors' terms in reviewing the research literature (e.g., "challenge," "growth mindset"), these studies all address individuals' perceptions of difficulty.

Our conceptualization of perceived difficulty draws on coping, flow, and mindset research. Perceived challenge is part of the appraisal process in Lazarus (1991) coping model, whereby challenge leads to positive emotions. Within flow theory, challenge positively motivates engagement and discourages boredom, when balanced with skill and interest (Csíkszentmihályi 1990, 1988; Csíkszentmihályi and Schneider 2000). Moreover, encountering challenge has been found to promote women's persistence in computer science (Milesi et al. 2017). However, girls who do not have a growth mindset, i.e., the belief that mathematics ability can be developed rather than innate, may struggle with stereotypes about their intelligence when tasked with difficult mathematics problems (Dweck 2000, 2006). Recent studies have shown a negative association between women's and African Americans' participation in STEM fields and the overall perception of those fields as requiring "brilliance," (Meyer et al. 2015; Leslie et al. 2015). Overall, previous theory suggests that challenging or difficult material or concepts can either motivate or discourage students within specific domains.

### 1.3. Present Study: Difficulty Orientations, Race/Ethnicity, and Gender

Given our interest in ability beliefs specific to difficult academic work, we focus this study on the relationship between students' difficulty orientation and PEMC outcomes, specifically, declared major and completed degree field. In a previous study, we confirmed that high school men and women had significantly different perceived ability regarding the level of mathematics challenge, and that these perceptions were related to advanced science course-taking, intended major, and major declared (Nix et al. 2015). We refined the scales since the earlier study and here refer to these ability beliefs as *difficulty orientations*, uniquely focusing on challenging academic work, especially in mathematics. Notably, in this study we compare domain-specific (i.e., mathematics, verbal) and domain-general measures. Using a gender and race/ethnicity intersectionality approach, we seek to establish whether difficulty orientations in general, verbal, and mathematics domains significantly differ by race/ethnicity and gender categories.

## 2. Method

### 2.1. Data Source and Participants

This study uses the full panel of restricted-use Education Longitudinal Study (ELS) 2002/2012 data from the National Center for Education Statistics, including the Postsecondary Education Transcript Study (PETS). In 2002, a nationally representative sample of about 16,200 10th graders from about 750 high schools responded to the ELS base year survey. Because not all students make school and degree transitions "on time," especially underrepresented and lower-SES students, we refer to each time point by year rather than assuming completion of a degree milestone. Follow-up surveys were distributed in 2004 (students' 12th grade year), 2006 (two years after high school), and 2012 (eight years after high school) (Ingels et al. 2007).

We define our population of interest as 10th graders in 2002 who earned at least a bachelor's degree by 2012 (*n* = 11,535). While missing data is not surprising with a national longitudinal dataset spanning ten years on students' educational and career transitions, we used multiple imputation to address this issue (Cox et al. 2014; Rubin 2004).[1] Specifically, we used the Monte Carlo chained equation method built into Stata 14 to generate 10 imputed datasets after conducting 100 imputations for each dataset (see Klein 2016). Across our analyses, we produced robust pooled estimates through Stata 14's *mi estimate* commands. We also used NCES-provided panel weight *f3bypnlpswt* during both the multiple imputation and analyses to more accurately represent the national population.

### 2.2. Measures

**Dependent Variables.** Dependent variables include participants' *declared major* in 2006, two years after high school and first completed *degree major* field as of 2012, eight years after high school. Majors are coded to compare mathematics-intensive PEMC fields (physical sciences, engineering, mathematics, and computer sciences) with other STEM (biological sciences, health sciences, and social/behavioral and other sciences) and non-STEM fields, which serve as the reference group. Declared major includes an undeclared/undecided category to capture students who had not yet selected a field of study or who had delayed entry into postsecondary education.

**Independent Variables: Difficulty Orientations.** Questionnaires in students' 10th grade year included Likert-scale items regarding perceived ability to learn the most "difficult," "hard," or "complex" material in general as well as in English or mathematics classes. These items were originally developed for self-efficacy scales in PISA:2000 and modified for ELS:2002 (Ingels et al. 2004; OECD n.d.). However, given our interest in difficulty orientations, we focused our analyses on the six items that measure students' perceptions of their own abilities with difficult or challenging material. After developing our 10 datasets using multiple imputation, we used confirmatory factor analysis to develop three scales that reflect students' difficulty orientation by domain: *general difficulty orientation* (alpha = 0.7), *verbal difficulty orientation* (alpha = 0.9), and *mathematics difficulty orientation* (alpha = 0.9). Table A1 provides a description of the items for each scale, factor loadings, scoring coefficients, eigenvalues, and average alpha coefficients, all which meet generally acceptable levels for usage as scales (Kline 2011).

**Covariates.** Given the above-cited prominence of background and educational experiences in the literature, the analysis additionally included demographic factors (*gender, race/ethnicity, family income, and parent education*), high school experiences (*standardized test scores, science course taking, GPA,*

---

[1]  A total of 9315 or about 81% of observations were missing data on at least one of the 24 independent or dependent variables in the model. There are 11,535 cases total in the dataset. For example: Sex = 0 missing, Race = 0 missing, Scipip = 944 missing, GPA12 = 958 missing, Growth = 2971 missing, Mathtxt10 = 3093 missing, Maj2006 = 4466 missing, and Majdeg = 5872 missing. Further detail can by supplied by the authors upon request.

*valuing mathematics*[2], and *mathematics growth mindset*[3]), high school characteristics (*percentage free/reduced lunch, region,* and *urbanicity*), postsecondary participation in *undergraduate research* with a faculty member, and postsecondary institutional characteristics (*control* and *selectivity* of the first attended institution). Table A2 shows pooled sample descriptive statistics for each of the covariates listed.

*2.3. Analysis*

This study examines the extent to which difficulty orientations, gender, and race/ethnicity predict mathematics-intensive degrees, independently and interdependently. The following research questions guide our study.

RQ1. Do domain-specific and domain-general difficulty orientation measures differ by gender and race/ethnicity identity categories?

**H1A.** *High school boys report higher difficulty orientations than their female peers, particularly in mathematics. In other words, boys' mathematics difficulty orientation scores will be higher than girls'.*

**H1B.** *Non-White students' difficulty orientations will be lower than those of their White peers.*

RQ2. To what extent do difficulty orientation measures predict PEMC degrees?

**H2.** *Students with higher mathematics difficulty orientations will be more likely to declare PEMC majors and earn PEMC degrees, all else being equal.*

RQ3. Do the relationships between difficulty orientation and PEMC degrees differ by gender and race/ethnicity?

**H3.** *The relationship between mathematics difficulty orientation and PEMC outcomes will be greater among non-White students than White students and among women than men, such that the relationship for White men will be weaker than for other gender and race/ethnicity groups.*

To answer the first research question, we estimated linear regression models to evaluate how difficulty orientation differs by gender and race/ethnicity.[4] To address the second research question, we estimated a series of multinomial logistic regression models, progressively introducing difficulty orientation measures to estimate their effects on declared/degree major, while controlling for the covariates listed in the previous section. While our reporting focuses on results for PEMC fields, the models carefully consider gradations in declared/degree majors rather than a binary PEMC/non-PEMC model. Non-STEM majors serve as the reference group, as compared to (a) PEMC; (b) other STEM, and in the declared major models; and (c) undeclared/undecided majors.

We started with a base model (Equation (1)), including the dependent variable of interest, gender, race/ethnicity, and control variables.

$$mlogit(major) = \beta_0 + \beta_1 gender + \beta_2 race + \beta_3 S + \beta_4 HS + \beta_4 research + \beta_5 PSI + u \qquad (1)$$

where

*major* = declared or degree major (see "Dependent Variables" section);

*S* = student-level controls (family income, parent education standardized test scores, science course taking, GPA, mathematics value, and growth mindset);

---

[2] The "valuing mathematics" item asked participants of the 10th grade ELS survey about their agreement with the statement, "Mathematics is important to me personally."

[3] The 10th grade ELS survey included an item asking participants about their agreement with the statement, "Most people can learn to be good at math," (Ingels et al. 2007). We have labeled this item "growth mindset" given its relationship with Dweck (2000, 2006) construct.

[4] Traditionally, we would use mean-item *t*-tests and one-way analysis of variance tests to address this question. However, Stata 14 does not allow the estimation of these statistics using multiply-imputed data.

*HS* = high school characteristics (percentage free and reduced lunch, region, and urbanicity);

*research* = participation in undergraduate research; and

*PSI* = postsecondary institutional characteristics (control and selectivity of the first attended institution)

To capture the domain-specific effects of each difficulty orientation, we estimated four additional models. The first three added only one of the difficulty orientations to the base model (Equations (2)–(4)). The last model in this sequence included all three of the difficulty orientation scales (Equation (5)).

$$mlogit(major) = \beta_0 + \beta_1 gender + \beta_2 race + \boldsymbol{\beta_3 general} + \beta_4 S + \beta_5 HS \\ + \beta_6 research + \beta_7 PSI + u \tag{2}$$

$$mlogit(major) = \beta_0 + \beta_1 gender + \beta_2 race + \boldsymbol{\beta_3 verbal} + \beta_4 S + \beta_5 HS \\ + \beta_6 research + \beta_7 PSI + u \tag{3}$$

$$mlogit(major) = \beta_0 + \beta_1 gender + \beta_2 race + \boldsymbol{\beta_3 math} + \beta_4 S + \beta_5 HS + \beta_6 research \\ + \beta_7 PSI + u \tag{4}$$

$$mlogit(major) = \beta_0 + \beta_1 gender + \beta_2 race + \boldsymbol{\beta_3 general} + \boldsymbol{\beta_4 verbal} \\ + \boldsymbol{\beta_5 math} + \beta_6 S + \beta_7 HS + \beta_8 research + \beta_9 PSI + u \tag{5}$$

where

*general* = domain-general difficulty orientation scale,

*verbal* = verbal difficulty orientation scale, and

*math* = mathematics difficulty orientation scale.

Our final research question (RQ3) examines whether the relationship between difficulty orientations and PEMC outcomes varies by gender and race/ethnicity. In our preliminary analyses, we tested for significant differences in gender and race/ethnicity slopes by including interaction terms.[5] Because these interaction terms were statistically insignificant in our initial model results, they were therefore removed from our final models, and they are not shown in our mathematical expressions of these models above. Despite the null findings for the interaction terms, we hypothesized there could still be meaningful differences in the relationship between difficulty orientations and PEMC outcomes by identity group.

Using the Equation (5) model, we used multinomial logistic regression (mlogit) models to predict students' PEMC major outcomes (declared and degree field). To better understand potential differences by race/ethnicity and gender, and to simplify interpretation of our results, we report these results as predicted probabilities. Post-estimation predicted probabilities were generated by *mimrgns*, a user-written Stata command that correctly produces pooled estimates of multiply-imputed data using Stata's built-in *margins* command and by applying Rubin's rules (Klein 2016). These predicted probabilities were estimated holding all other variables in Equation (5) constant.

First, we generated predicted probabilities to declare a major or earn a degree in PEMC by both gender and race/ethnicity for the 10th, 25th, 50th, 75th, and 90th percentiles of each difficulty orientation scale. Next, using the *pwcompare* option, we evaluated the statistical significance of differences in students' predicted probabilities of PEMC majors and degrees, by difficulty orientation, gender, and race/ethnicity. We assessed intersectional differences by identity (gender and race/ethnicity) as follows: (1) comparing women and men within race/ethnicity groups (e.g., Latinas vs. Latinos) and (2) race/ethnicity groups within gender categories (e.g., Latinos vs. White men). Finally, we examined the degree to which

---

[5]   Specifically, we included the following two-way cross-product terms separately in the model shown on Equation (5): (a) gender × race/ethnicity, (b) gender × math, and (c) race × math. We also tested a three-way interaction model by including gender × race × math with its corresponding two-way conditional effects in the model shown on Equation (5).

each identity group increased in percentile difficulty orientation. For instance, we tested whether the probability for Latinas at the 25th percentile differed from the probability for Latinas at the 10th percentile. Together, these results provide insights on the manner that PEMC outcomes are related to intersections between gender, race/ethnicity, and difficulty orientations.

## 3. Results

### 3.1. Descriptive Statistics

Descriptive statistics for the sample are shown in Appendix A Table A1 (difficulty orientation scale descriptions), Table A2 (covariate descriptive statistics), Table A3 (declared and degree major by sex), and Table A4 (declared and degree major by race/ethnicity). In brief, these statistics show that the sample is gender-balanced (48.4% women and 51.6% men); majority White (63.8%), majority middle income (52.6% from families earning $25,001–$75,000 per year); minority advanced science course takers (20.5% completed both a second chemistry and second physics courses in high school); and majority public college attendees (76.6%) (Table A2). PEMC ranks third for men's declared and degree major (14.4% and 13.6%, respectively), but last for women's declared and degree major (3.7% and 3.6%, respectively) (Table A3). Asian/Pacific Islander and Black students are more likely than White students to declare PEMC majors (12.7% and 11.2% vs. 8.6%), but Asian/Pacific Islanders are the only group more likely to earn PEMC degrees than White students (12.7% vs. 8.7%) (Table A4). Black students' rate of PEMC participation drops from 11.2% two years after high school to 8.1% eight years after high school. These statistics help frame our study and show meaningful variation, particularly between gender and racial/ethnic groups in pursuit of PEMC majors and degrees.

### 3.2. RQ1: Do Difficulty Orientation Measures Differ by Gender and Race/Ethnicity?

This study is chiefly interested in estimating the relationship between undergraduate PEMC outcomes and the following, potentially intersecting predictors: difficulty orientations, gender, and race/ethnicity. We found pronounced differences in mean difficulty orientations between men and women, but more variable differences between White and non-White students. High school boys and girls vary in their mathematics difficulty orientation. Where boys on average scored 0.2 SD above the mean in their orientation towards difficult mathematics, girls scored 0.2 SD below the mean (Table 1; $p < 0.001$). Girls and boys did not vary significantly on their general or verbal difficulty orientations, indicating this is a domain-specific difference.

**Table 1.** Difficulty Orientations by Gender.

|  | Men | | Women | | | Range | |
| --- | --- | --- | --- | --- | --- | --- | --- |
|  | **Mean** | **SE** | **Mean** | **SE** | **Sig.** | **Min** | **Max** |
| General Academic Scale | 0.0 | 0.0 | 0.0 | 0.0 | | −1.7 | 1.1 |
| Verbal Scale | 0.0 | 0.0 | 0.0 | 0.0 | | −1.7 | 1.4 |
| Mathematics Scale | 0.2 | 0.0 | −0.2 | 0.0 | *** | −1.4 | 1.5 |

Note: $n$ = 11,535 respondents from the National Center for Education Statistics' (NCES) Education Longitudinal Study 2002/2012 restricted data. Means and standard errors are reported. Restricted-use NCES data required rounding these descriptive results to the nearest tenth. Scales were developed using factor analysis, which automatically standardizes them to mean = 0 and SD = 1. * $p < 0.05$, ** $p < 0.01$, *** $p < 0.001$.

Turning to differences by race/ethnicity, Table 2 shows Latino students are the only group with lower mathematics difficulty orientation compared to White students; in fact, Latino students rated themselves at least 0.1 standard deviations lower than White students across all three difficulty orientations (Table 2; all $p < 0.01$). By contrast, Asian students' mathematics difficulty orientation is 0.2 SDs above White students, on average ($p < 0.01$). Notably, Black and other race/ethnicity students' difficulty orientation scores were not significantly different from those of White students.

**Table 2.** Difficulty Orientations by Race/Ethnicity.

| | White (Reference) | | Asian/Pacific Islander | | | Black | | | Latino | | | Other Groups | | | Range | |
|---|---|---|---|---|---|---|---|---|---|---|---|---|---|---|---|---|
| | Mean | SE | Mean | SE | Sig. | Mean | SE | Sig. | Mean | SE | Sig. | Mean | SE | Sig. | Min | Max |
| General Academic Scale | 0.0 | 0.0 | 0.1 | 0.0 | | 0.0 | 0.0 | | −0.1 | 0.0 | ** | 0.0 | 0.1 | | −1.7 | 1.1 |
| Verbal Scale | 0.1 | 0.0 | 0.0 | 0.0 | | 0.0 | 0.0 | | −0.1 | 0.0 | ** | 0.0 | 0.1 | | −1.7 | 1.4 |
| Mathematics Scale | 0.0 | 0.0 | 0.2 | 0.0 | ** | 0.0 | 0.0 | | −0.1 | 0.0 | ** | 0.0 | 0.1 | | −1.4 | 1.5 |

Note: $n$ = 11,535 respondents from the National Center for Education Statistics' (NCES) Education Longitudinal Study 2002/2012 restricted data. Restricted-use NCES data required rounding these descriptive results to the nearest tenth. Scales were developed using factor analysis, which automatically standardizes them to mean = 0 and SD = 1. Significance levels are produced comparing against means on White. * $p < 0.05$, ** $p < 0.01$, *** $p < 0.001$.

### 3.3. RQ2: Do Difficulty Orientations Predict Mathematics-Intensive Majors and Degrees?

We next estimated multinomial logistic regression models, progressively introducing difficulty orientation measures to evaluate their relative effects on majors and degrees earned in PEMC fields. Because of space constraints, we report findings for only one category of the outcome variables in these models: PEMC declared and degree major. Tables displaying results for undeclared/undecided, biological sciences, health sciences, and social/behavioral and other sciences declared and degree majors are available upon request. We report the difficulty orientation results as relative risk ratios (RRRs), where ratios lower than 1 are interpreted as 1 minus the relative risk ratio.[6]

Table 3 reports findings for declared majors two years after high school (four years after 10th grade students' difficulty orientations were measured). In the models with only one difficulty orientation (Equations (2)–(4)), only beliefs about mathematics ability reached significance: a one standard deviation increase in mathematics difficulty orientation predicted a 34% increase in the risk of declaring PEMC versus non-STEM majors ($p < 0.01$). In the full model with all three difficulty orientations (Equation (5)), both verbal and mathematics domains mattered. Verbal difficulty orientation was negatively associated with PEMC; a positive standard deviation change decreased the risk of majoring in these fields by 24% (RRR = 0.76; $p < 0.01$). By contrast, mathematics difficulty orientation predicted a 49% *increase* in the risk of declaring PEMC versus non-STEM majors ($p < 0.001$).

Table 4 reports the findings for degree field. In the single difficulty orientation models here (Equations (2)–(4)), only the verbal domain emerged as significant (RRR = 0.79; $p < 0.05$), in a negative direction as in the declared major model from Table 3. When accounting for all difficulty orientations (Equation (5)), both verbal and mathematics difficulty orientations were again significantly associated with PEMC, in opposite directions. Specifically, a one standard deviation increase in verbal difficulty orientation was associated with a 28% (RRR = 0.72; $p < 0.01$) decrease in the risk of earning a PEMC degree versus a non-STEM degree. Again, mathematics difficulty orientation was positively associated with PEMC outcomes, whereby a one standard deviation increase predicted a 38% increased risk of earning a PEMC degree ($p < 0.001$).

In summary then, domain-specific difficulty orientations were more influential than domain-general difficulty orientations. To answer the question, do difficulty orientations predict PEMC outcomes: yes, verbal and especially mathematics difficulty orientations do have significant effects on students' chances of declaring PEMC majors and earning PEMC degrees, in distinct directions. Beliefs about one's ability with difficult verbal tasks were negatively associated with PEMC. Conversely, beliefs about difficult mathematics were positively related to PEMC. Moreover, difficulty orientations measured in 10th grade had stronger effects on declared majors than on degree field, a later event. Next, our study adds complexity to these findings focused on the relationship between ability beliefs and postsecondary outcomes, from high school through college. The final research question focuses on intersections.

---

[6]　Following the expression of the results in relative risk ratios (RRRs), we use the term "risk" regardless of the positive or negative connotation of the outcome. RRRs require the use of "risk" over other terms because they measure the likelihood of occurrences in one group compared to the likelihood of occurrences in other groups, rather than the likelihood of occurrences versus non-occurrences as is the case of odds ratios (Andrade 2015).

**Table 3.** PEMC Major Declared Two Years after High School, by Sex, Race/Ethnicity, and Difficulty Orientations (D.O.).

| | Base Model | | | Base + General | | | Base + Verbal | | | Base + Math | | | Base + All D.O. | | |
|---|---|---|---|---|---|---|---|---|---|---|---|---|---|---|---|
| | PP | RRR | SE | PP | RRR | SE | PP | RRR | SE | PP | RRR | SE | PP | RRR | SE |
| *Demographic Characteristics* | | | | | | | | | | | | | | | |
| Sex | | | | | | | | | | | | | | | |
| Male (Reference) | 13.88% | - | - | 13.89% | - | - | 13.95% | - | - | 13.49% | - | - | 13.43% | - | - |
| Female | 3.89% | 0.23 *** | 0.03 | 3.89% | 0.23 *** | 0.03 | 3.87% | 0.22 *** | 0.03 | 4.03% | 0.24 *** | 0.03 | 4.05% | 0.24 *** | 0.04 |
| Race/Ethnicity | | | | | | | | | | | | | | | |
| White (Reference) | 8.13% | - | - | 8.13% | - | - | 8.12% | - | - | 8.16% | - | - | 8.14% | - | - |
| Asian/Pacific Islander | 8.35% | 1.22 | 0.25 | 8.33% | 1.22 | 0.25 | 8.21% | 1.20 | 0.25 | 8.53% | 1.244 | 0.25 | 8.41% | 1.21 | 0.25 |
| Black | 13.60% | 2.12 *** | 0.41 | 13.63% | 2.12 *** | 0.41 | 13.83% | 2.18 *** | 0.42 | 13.34% | 2.06 *** | 0.40 | 13.65% | 2.13 *** | 0.41 |
| Latino | 8.65% | 1.22 | 0.28 | 8.67% | 1.22 | 0.28 | 8.69% | 1.23 | 0.28 | 8.53% | 1.19 | 0.27 | 8.54% | 1.19 | 0.27 |
| Other | 9.02% | 1.17 | 0.39 | 9.02% | 1.17 | 0.39 | 9.03% | 1.18 | 0.39 | 8.95% | 1.16 | 0.38 | 8.91% | 1.14 | 0.38 |
| *Difficulty Orientations* | | | | | | | | | | | | | | | |
| General Academic Scale | | | | | 0.99 | 0.13 | | | | | | | | 0.95 | 0.13 |
| Verbal Scale | | | | | | | | 0.85 | 0.08 | | | | | 0.76 ** | 0.07 |
| Mathematics Scale | | | | | | | | | | | 1.34 ** | 0.13 | | 1.49 *** | 0.15 |
| Constant | | 0.00 *** | 0.00 | | 0.00 *** | 0.00 | | 0.00 *** | 0.00 | | 0.01 *** | 0.01 | | 0.01 *** | 0.01 |
| *f*-statistic | 7.26 *** | | | 7.08 *** | | | 7.18 *** | | | 6.95 *** | | | 6.65 *** | | |
| Observations | 11,535 | | | 11,535 | | | 11,535 | | | 11,535 | | | 11,535 | | |

Note: $n = 11,535$ respondents from the National Center for Education Statistics' Education Longitudinal Study 2002/2012 restricted data. Parent education, family income, 10th grade standardized test scores, science course taking, high school GPA, mathematics value, mathematics growth mindset, percentage free and reduced-price lunch, high school region, high school urbanicity, participation in undergraduate research, institutional control, and college selectivity was included in the model, but not shown for space. Full table is available upon request. * $p < 0.05$, ** $p < 0.01$, *** $p < 0.001$.

**Table 4.** PEMC Completed Degree Field, by Sex, Race/Ethnicity, and Difficulty Orientations (D.O.).

| | Base Model | | | Base + General | | | Base + Verbal | | | Base + Math | | | Base + All D.O. | | |
|---|---|---|---|---|---|---|---|---|---|---|---|---|---|---|---|
| | PP | RRR | SE | PP | RRR | SE | PP | RRR | SE | PP | RRR | SE | PP | RRR | SE |
| *Demographic Characteristics* | | | | | | | | | | | | | | | |
| Sex | | | | | | | | | | | | | | | |
| Male (Reference) | 12.86% | | | 12.89% | | | 12.94% | | | 12.66% | | | 12.60% | | |
| Female | 3.88% | 0.27 *** | 0.05 | 3.87% | 0.27 *** | 0.05 | 3.85% | 0.26 *** | 0.05 | 3.95% | 0.28 *** | 0.05 | 3.97% | 0.28 *** | 0.05 |
| Race/Ethnicity | | | | | | | | | | | | | | | |
| White (Reference) | 8.22% | - | - | 8.21% | - | - | 8.19% | - | - | 8.23% | - | - | 8.21% | - | - |
| Asian/Pacific Islander | 8.10% | 1.16 | 0.24 | 8.05% | 1.15 | 0.23 | 7.92% | 1.13 | 0.23 | 8.23% | 1.18 | 0.24 | 8.12% | 1.16 | 0.24 |
| Black | 10.78% | 1.56 | 0.36 | 10.86% | 1.57 | 0.36 | 11.08% | 1.63 * | 0.38 | 10.65% | 1.53 | 0.36 | 10.98% | 1.60 * | 0.38 |
| Latino | 8.52% | 1.18 | 0.32 | 8.54% | 1.18 | 0.32 | 8.58% | 1.19 | 0.33 | 8.45% | 1.16 | 0.32 | 8.45% | 1.16 | 0.33 |
| Other | 6.93% | 0.91 | 0.33 | 6.92% | 0.91 | 0.33 | 6.92% | 0.91 | 0.34 | 6.90% | 0.90 | 0.33 | 6.82% | 0.89 | 0.34 |
| *Difficulty Orientations* | | | | | | | | | | | | | | | |
| General Academic Scale | | | | | 0.94 | 0.11 | | | | | | | | 0.98 | 0.14 |
| Verbal Scale | | | | | | | | 0.79 * | 0.09 | | | | | 0.72 ** | 0.09 |
| Mathematics Scale | | | | | | | | | | | 1.217 | 0.15 | | 1.38 * | 0.18 |
| Constant | | 0.00 *** | 0.00 | | 0.00 *** | 0.00 | | 0.00 *** | 0.00 | | 0.00 *** | 0.00 | | 0.00 *** | 0.00 |
| *f*-statistic | 5.48 *** | | | 5.31 *** | | | 5.23 *** | | | 5.29 *** | | | 4.78 *** | | |
| Observations | 11,535 | | | 11,535 | | | 11,535 | | | 11,535 | | | 11,535 | | |

Note: *n* = 11,535 respondents from the National Center for Education Statistics' Education Longitudinal Study 2002/2012 restricted data. Parent education, family income, 10th grade standardized test scores, science course taking, high school GPA, mathematics value, mathematics growth mindset, percentage free and reduced-price lunch, high school region, high school urbanicity, participation in undergraduate research, institutional control, and college selectivity was included in the model, but not shown for space. Full table is available upon request. * $p < 0.05$, ** $p < 0.01$, *** $p < 0.001$.

### 3.4. RQ3: Do the Relationships between Difficulty Orientations and PEMC Outcomes Vary by Gender and Race/Ethnicity?

We begin with the identity characteristics reported in the models we just reviewed in Tables 3 and 4. In the base models, women have a 3.9% predicted probability of declaring a PEMC major and earning a PEMC degree ($p < 0.001$). Across the models, the negative relationship between female gender and PEMC outcomes persists. When the mathematics measure is added to the base model (base + math), women see a 0.14 and 0.07 percentage point gain in their probability to declare and earn a degree in PEMC, respectively. An additional 0.02 percentage point was gained when the other two difficulty orientation measures are added in the full model (Equation (5)).

Turning to race/ethnicity, our multinomial regression analysis yielded significant findings for only one group: Black students. In the base model, Black students have a 13.6% probability of majoring in PEMC fields, 5.47 percentage points higher than White students ($p < 0.001$). We found a modest increase in Black students' predicted probability of declaring PEMC majors in the base + verbal model (13.8%), but a decrease in the base + math model (13.3%). This counterintuitive finding suggests Black students are more likely to declare PEMC majors as their perceived ability to learn difficult *verbal* material increases, but mathematics difficulty orientation may be related to pursuing non-STEM careers rather than PEMC. In the degree completion models, the association between being Black and earning PEMC degrees appears primarily associated with verbal ability beliefs, which again increased students' probability of earning PEMC degrees to 11.1% in the base + verbal model and 11.0% in the full model. These models show specific gender and race/ethnicity effects. Next, we attend more closely to their intersecting effects on PEMC degree outcomes.

**Intersectional analyses.** First, we estimated multinomial logistic regression models with two-way and three-way interaction terms. There were no significant findings on the interaction terms (see Table A5), indicating that the relationship between PEMC outcomes and mathematics difficulty orientations generally moves in the same direction for all categories of gender, race/ethnicity, and gender and race/ethnicity together. While there were no significant slope differences, predicted probabilities provide the opportunity to examine nuanced differences by race/ethnicity and gender. Three notable overarching results emerged from our analysis of the predicted probabilities. First, women of every race/ethnicity had a lower probability than men of every race/ethnicity to declare PEMC majors and earn PEMC degrees. Second, Black men and women had higher probabilities to declare PEMC majors and earn PEMC degrees as compared to their White peers. Third, for all identity groups, each percentile increase in mathematics difficulty orientation is associated with a significant increase in probability to declare PEMC majors; this is not the case for earning PEMC degrees.

*Gender*. We find the gender disparity observed above persists at all levels of mathematics difficulty orientation, irrespective of race/ethnicity. Figures 1 and 2 most immediately illustrate the differences between men and women's probability to declare or earn a degree in PEMC, by race/ethnicity and mathematics difficulty orientation. Appendix A Figures A1 and A2 show the statistical significance levels of increases in the predicted probability of declaring PEMC majors, by gender, race/ethnicity, and difficulty orientation percentiles (10th–90th, consistent with Figures 1 and 2, all $p < 0.05$ or smaller). Men's predicted probabilities to declare PEMC majors ranged from 7.2% to 29.4% between the 10th and 90th percentiles (Figure 1). Women ranged from 2.0% to 10.2% (Figure 1). For degree major, men ranged from 6.9% to 21.5%, whereas women ranged from 2.0% to 7.3% (Figure 2).

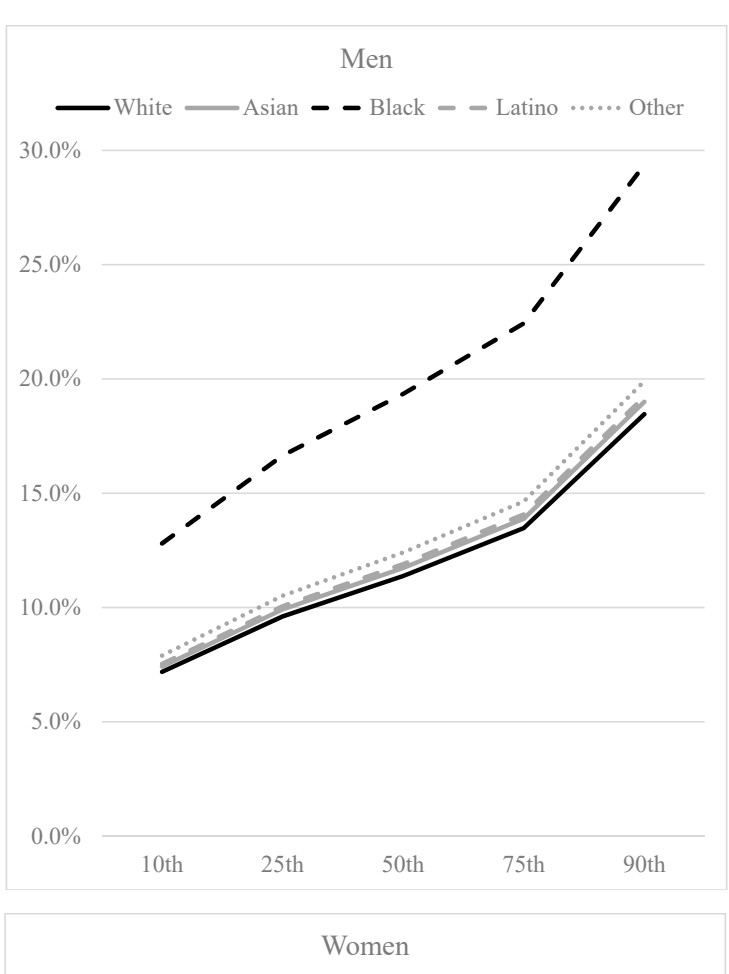

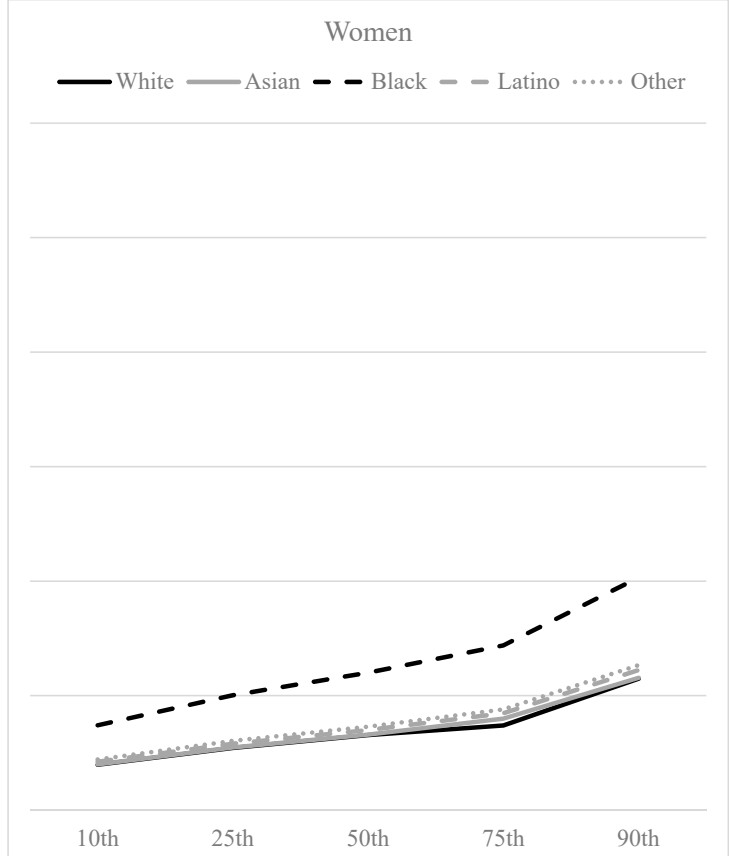

**Figure 1.** PEMC Declared Major Given Mathematics Difficulty Orientation Percentiles, Gender, and Race/Ethnicity.

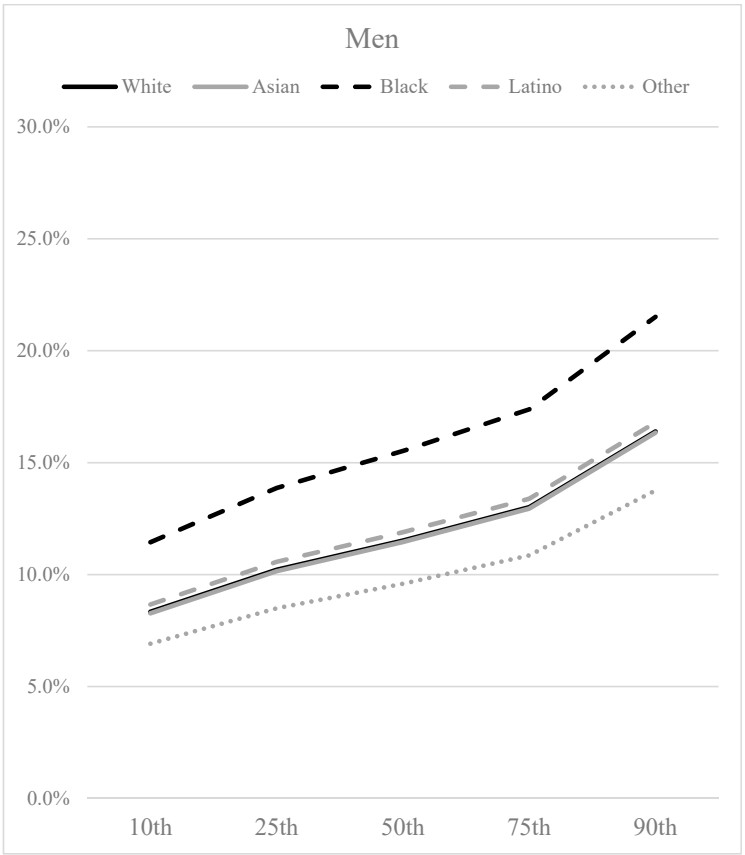

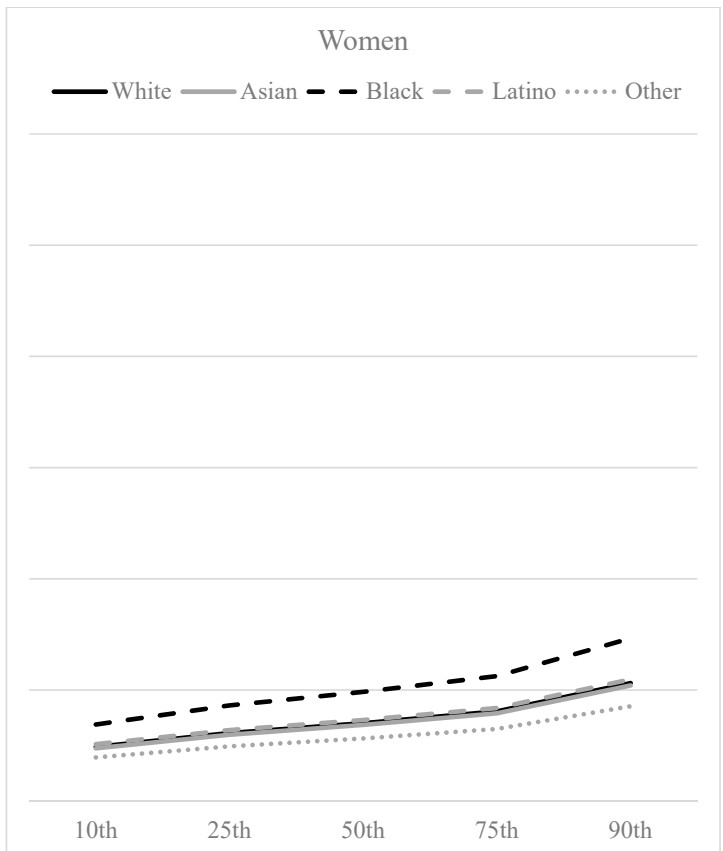

**Figure 2.** PEMC Degree Major Given Mathematics Difficulty Orientation Percentiles, Gender, and Race/Ethnicity.

***Race/Ethnicity***. Some gender findings were specific by race/ethnicity. Across both outcomes (declared major and degree major), Black men and women had the highest probability of participating in PEMC fields given their perceptions of their ability with difficult mathematics (Figures 1 and 2). This follows the significant differences by race/ethnicity only for Black students shown in earlier regression results, as compared with their White peers. For example, our model predicts Black women and men with mean-level mathematics difficulty orientation had a 6.0% and 19.3% probability of declaring a PEMC major, respectively, all else equal. By contrast, these predicted probabilities were nearly double that of their White peers: 3.3% for White women and 11.4% for White men.

Overall, Black students had a higher predicted probability of majoring in PEMC fields as compared to White students, no matter which difficulty orientation is integrated in the models (Table 3). However, they were not significantly more likely to earn PEMC degrees until accounting for verbal difficulty orientations (Table 4). Black men's and women's predicted probabilities of declaring PEMC majors ranged from 12.8 to 29.4% and 3.7 to 10.2% from the lowest to highest levels of mathematics difficulty orientations, respectively, versus White men's and women's probabilities, which range from 7.2 to 18.5% and 2.0 to 5.7%, respectively (Figure 1). For PEMC degrees, Black men and women's predicted probabilities ranged 11.4–21.5% and 3.5–7.3%, respectively, versus White men and women's predicted probabilities: 8.3–16.4% and 2.4–5.3% (Figure 1).

***Mathematics difficulty orientation***. We find significant gains in PEMC outcomes given increases in mathematics difficulty orientations. Appendix A Figures A1 and A2 show that for all race/ethnicity groups, there were significant gains in probability to declare a PEMC major when moving from the 10th to the 25th percentile, the 25th to the 50th percentile, the 50th to the 75th percentile, and the 75th to the 90th percentile. Figures A1 and A2 show the largest gains for all students occurred between the 75th and 90th percentiles. Gains for non-Black women hovered between 0.6 and 0.8 percentage points until the 75th to 90th percentile, when they see a 1.8–1.9 percentage point jump in probability to declare a PEMC major (Figure A2). For non-Black men, gains from the 10th to the 75th percentiles hovered around 1.8–2.6 percentage points, and then jumped to 5.0–5.3 percentage points between the 75th and 90th percentiles (Figure A1). There were no significant gains in these percentile changes in probability to earn PEMC degrees.

***Gender and race/ethnicity revisited***. As mathematics difficulty orientation increases, Black men and women experienced the largest gains in their PEMC outcomes. As mathematics difficulty orientation rose, increases in predicted probabilities for White, Asian, Latino, and other race/ethnicity men fell between 0.8 and 1.6 percentage points below Black men's increase in probability to declare PEMC majors (Figure A1). Similarly, predicted probabilities of declaring PEMC majors among White, Asian, Latino, and other race/ethnicity women fell between 0.4 and 1.1 percentage points below that of Black women (Figure A2). Moreover, PEMC degree major had narrower sex differences between men's and women's predicted probabilities, attributable in part to the considerable drop in men at the 90th percentile between declaring PEMC majors and completing these degrees; this was true for Black men in particular. Almost a third (29.0%) of Black degree earners at the 90th percentile of mathematics difficulty orientation declared PEMC majors (Figure 1), but only 21.5% earn degrees in this field (Figure 2).

## 4. Discussion

### 4.1. Summarizing and Contextualizing Findings

This nationally representative longitudinal study examines intersections in the nuanced relationships between mathematics difficulty orientation, gender, race/ethnicity on mathematics-intensive PEMC majors and degrees. We asked the following questions and answer them succinctly, and then with more detail, below. (1) Do domain-specific and domain-general difficulty orientation measures differ by gender and race/ethnicity identity categories? Yes, mathematics difficulty orientation varies by both gender and, less consistently, race/ethnicity. (2) To what extent does difficulty orientation predict selection and degree

attainment of PEMC majors? Mathematics difficulty orientations positively predict PEMC outcomes, holding all else constant; verbal difficulty orientation negatively predicts PEMC degrees. (3) Do the relationships between difficulty orientation and PEMC outcomes differ by gender and race/ethnicity categories? While the direction does not differ, the magnitude does, with particularly intriguing findings for Black men and women.

Boys' ability beliefs about difficult mathematics, i.e., mathematics difficulty orientations, is higher than that of girls, as has been observed in other studies (Nix et al. 2015; Perez-Felkner et al. 2017). As compared to White students, Latino students had lower mathematics difficulty orientation, and Asian/Pacific Islanders had higher mathematics difficulty orientations. Latino students also differed from White students on the verbal and general measures. Mathematics and verbal difficulty orientations also predicted PEMC outcomes. Mathematics difficulty orientation positively predicted declaring PEMC majors independently and in combination with the full set of difficulty orientation measures. With respect to PEMC degrees, verbal difficulty orientation had a negative effect, both in the base + verbal model and in the model with all three difficulty orientation measures. Mathematics difficulty orientation had a positive effect on PEMC degrees, but only in the model with all three difficulty orientation measures.

The magnitude of the relationships between mathematics difficulty orientations and PEMC outcomes shifted given gender and race/ethnicity identity categories. Still, differences were largest and most consistent between men and women, indicating gender more strongly predicts PEMC postsecondary outcomes than race/ethnicity. Women were less likely than men to declare a PEMC major or earn a PEMC degree. This finding is consistent with previous research showing lower ability beliefs among girls and women as compared to boys and men (Beyer 1990; Beyer and Bowden 1997), especially in mathematics and science domains (Correll 2001; Sax 1994).

Black students were more likely to declare PEMC majors and earn these degrees than White students (Tables 3 and 4), and indeed all other students (Figures 1 and 2). Notwithstanding, Black men and women showed higher than expected gains in probability to declare a PEMC major compared to their White, Latino, Asian, and other race/ethnicity counterparts when controlling for mathematics difficulty orientations and a host of background variables. Yet, Black students, especially Black women, are underrepresented in mathematics-intensive science fields (Anderson and Kim 2006; Ong et al. 2011). This suggests that factors beyond those measured, such as structural racism, act as barriers to these students' participation in STEM fields at expected rates (McGee and Bentley 2017; Smith and Gayles 2018). Our analyses that included interactions between gender, race/ethnicity, and difficulty orientations did not yield significant results, suggesting that difficulty orientations do not change the direction of the relationships between identity group and PEMC major and degree. However, our close look at predicted probabilities by gender and race/ethnicity group does show that the magnitude of the relationship between mathematics difficulty orientation and PEMC participation varies by identity group. This finding holds promise for the role that educators and institutions can play in encouraging underrepresented groups to select and complete mathematics-intensive science degrees.

## 4.2. Implications

**For postsecondary policy and practice.** High school and college faculty, advisors, and program coordinators are well-positioned to influence women and underrepresented students' beliefs in their ability to learn difficult mathematics material and invite them to engage in research opportunities, potentially increasing their persistence in PEMC fields (see also Espinosa and Nellum 2015). University administrators and institutional researchers can leverage the data to investigate coursework patterns among women and students of color, to illuminate potential policy changes in advising and major mapping practices. Mathematics difficulty orientations may indicate openness to more advanced learning experiences, including undergraduate research, ideally paired with supportive mentorship that continues to bolster students' confidence.

Practitioners and policy makers may also consider emphasizing the verbal and creative qualities of college-level science coursework, to attract students with confidence in those areas (Sax et al. 2017).

This may prove useful for Latinos as well, given the findings reported above on their lower difficulty orientations. STEM readiness among Latino students has been identified as a problem (Gandara 2006). While few studies have focused closely on Latino STEM undergraduates, a study by Cole and Espinoza (2008) using CIRP data found Latinas outperform Latinos in STEM college classrooms.

**For research and scholarship.** When controlling for all independent variables, mathematics difficulty orientations were positively associated with both declaring and earning a degree in PEMC fields, but verbal difficulty orientations were negatively associated. This finding is congruent with self-concept research showing that ability beliefs are domain-specific (Marsh 1986; Guay et al. 2003). Moreover, students with high verbal and mathematics difficulty orientations may not participate in PEMC because they perceive that they have greater choice in degree field, and see non-PEMC fields as more attractive, as has been found in studies on gender differences (Denissen et al. 2007; Wang et al. 2013).

Given Black college degree-earners' probability of earning PEMC degrees is lower than their probability of declaring these majors, all else equal, it remains an important question: what degree fields do they complete, and why do they switch out of these majors? In a qualitative study of prospective STEM majors at seven campuses in the early 1990s, Seymour (1999) found that, with the exception of the most socioeconomically disadvantaged, women who entered college as potential STEM majors were less rigid in their choice of major than were men. Perhaps in part explaining this result, Hanson (2008) finds that contemporary labor norms in the Black community contribute to their resilience, whereby female gender serves as an advantage for Black girls pursuing scientific careers. Recent research shows wider gender gaps among more affluent and more White school districts in the U.S. (Reardon et al. 2018); gender gaps also seem wider where inequality is smaller internationally, with upward mobility motivating women towards high-earning fields (Breda et al. 2018; Charles and Bradley 2009). Past studies have found Black girls report particularly high interest in science classes and careers (Hanson 2004; Riegle-Crumb et al. 2011). While there is limited research on the pathways of Black women in STEM fields, it is important to focus as well on Black men in STEM, who face both similar and distinct challenges in these fields (Lundy-Wagner 2013; Lundy-Wagner and Gasman 2011).

**Limitations and future directions**. With respect to the intersections between students' 10th grade difficulty orientations, gender, and racial/ethnic identities, we did not find significant interaction effects on postsecondary PEMC outcomes. It may be that women and students of color are more influenced by external actors and circumstances than their ability-related beliefs, given the importance of educational and social contexts on their STEM postsecondary outcomes (Charleston et al. 2014; Hurtado et al. 2011; Litzler et al. 2014; Ong et al. 2017). On the other hand, because our study included only a subsection of majors with the lowest participation of women, it is possible that we did not have a sufficient sample size to yield significant results via intersectional groups: non-White women who declared a PEMC major in the non-imputed dataset numbered less than 25 in each group.[7] Also, our difficulty orientation indicators were based on a limited number of factors. Future qualitative studies might further illuminate these groups' experiences and more richly detail the interplay between students' intersecting identities, perceived difficulty, and their perceived ability.

In leveraging this ten-year window to explain the pathways of women and racial/ethnic minorities from high school through undergraduate degree attainment in the span of a succinct manuscript, we did not focus this story on the covariates: high school and college characteristics and experiences. Notably however, previous studies highlight the importance of varied college experiences, especially for students of color (Chang et al. 2014; Cole and Espinoza 2008; Hurtado et al. 2011; Strayhorn et al. 2013). Our complete tables with all covariates (available by request) indicate undergraduate research participation positively predicts PEMC major declaration ($p < 0.01$) and degree completion ($p < 0.001$).

---

[7] Restricted-use data required that we round to the nearest 10 when reporting descriptive statistics to protect the identity of participants. Further, we report from the non-imputed dataset because multiple imputation can produce illogical results for dichotomous variables such as sex (Cox et al. 2014).

Future studies may provide particularly valuable insight into the relationship between undergraduate experiences, ability beliefs, and PEMC outcomes, for women and students of color in particular.

**Author Contributions:** S.N. led the writing and analysis of the manuscript, including the multiple imputation design and development of the tables and figures. L.P.-F. guided S.N. through data management and analysis design, suggested relevant literature, contributed substantive edits, and led reviewer responses.

**Funding:** This material is based upon work supported by the National Academy of Education/ Spencer Foundation Dissertation Fellowship, the P.E.O. Scholar Award, and National Science Foundation under Grant No. 1232139. Any opinions, findings, and conclusions or recommendations expressed in this material are those of the authors and do not necessarily reflect the views of the National Science Foundation. Article Processing Charges (APCs) were fully funded by institutions through the Knowledge Unlatched initiative.

**Acknowledgments:** Additional support was provided by Florida State University's Center for Postsecondary Success, Center for Higher Education Research, Teaching & Innovation, as well as doctoral research assistant Teng Zhao and undergraduate research assistants Abigail Smith and Madeline Ginn.

**Conflicts of Interest:** The authors declare no conflict of interest.

**Appendix A**

**Table A1.** Items, Factor Loadings, and Scoring Coefficients Used to Develop Difficulty Orientation Scales.

| Question | Factor Loadings | Scoring Coefficients |
|---|---|---|
| *General Difficulty Orientation* | | |
| *Eigenvalue = 2.8, Alpha coefficient = 0.7* | | |
| When I sit myself down to learn something really hard, I can learn it. | 0.6 | 0.1 |
| When studying, I keep working even if the material is difficult. | 0.8 | 0.2 |
| *Verbal Difficulty Orientation* | | |
| *Eigenvalue = 2.1, Alpha coefficient = 0.9* | | |
| I'm certain I can understand the most difficult material presented in English texts. | 0.8 | 0.3 |
| I'm confident I can understand the most complex material presented by my English teacher. | 0.9 | 0.4 |
| *Mathematics Difficulty Orientation* | | |
| *Eigenvalue = 2.2, Alpha coefficient = 0.9* | | |
| I'm certain I can understand the most difficult material presented in math texts. | 0.8 | 0.3 |
| I'm confident I can understand the most complex material presented by my math teacher. | 0.9 | 0.4 |

Note: $n$ = 11,535 respondents from the National Center for Education Statistics' Education Longitudinal Study 2002/2012 restricted data. Scales were estimated using factor analysis without rotation in Stata 14, which provides both factor loadings and scoring coefficients. All variables were loaded on a single factor with a minimum eigenvalue of 1.0. Alpha coefficients were at or above generally accepted levels (Kline 2011). Items were chosen based on its domain-specific expression of participants' perceived ability with challenging or difficult material. Participants were asked to indicate their level of agreement with each statement using a four-point Likert-scale.

**Table A2.** Sample Descriptive Statistics.

| | % or Mean | SE | Min | Max |
|---|---|---|---|---|
| *Demographic Characteristics* | | | | |
| Gender | | | | |
| Men | 48.4% | 0.7% | 0.0 | 100.0 |
| Women | 51.6% | 0.7% | 0.0 | 100.0 |
| Race/Ethnicity | | | | |
| White | 63.8% | 1.1% | 0.0 | 100.0 |
| Asian/Pacific Islander | 5.0% | 0.3% | 0.0 | 100.0 |
| Black | 13.0% | 0.7% | 0.0 | 100.0 |
| Latino | 13.6% | 0.8% | 0.0 | 100.0 |
| Other | 4.6% | 0.4% | 0.0 | 100.0 |

**Table A2.** Cont.

|  | % or Mean | SE | Min | Max |
|---|---|---|---|---|
| Parent Education |  |  |  |  |
| High School or Less | 21.1% | 0.8% | 0.0 | 100.0 |
| Some College | 31.9% | 0.8% | 0.0 | 100.0 |
| Bachelor's Degree | 28.1% | 0.8% | 0.0 | 100.0 |
| More Than a Bachelor's Degree | 18.9% | 0.7% | 0.0 | 100.0 |
| Family Income |  |  |  |  |
| Up to $25,000 | 16.6% | 0.7% | 0.0 | 100.0 |
| $25,001–$50,000 | 27.5% | 0.8% | 0.0 | 100.0 |
| $50,001–$75,000 | 25.1% | 0.7% | 0.0 | 100.0 |
| $75,001–$100,000 | 14.6% | 0.6% | 0.0 | 100.0 |
| $100,0001 or more | 16.3% | 0.7% | 0.0 | 100.0 |
| *High School Experiences* |  |  |  |  |
| 10th Grade Standardized Test Scores |  |  |  |  |
| Mathematics (mean) | 53.3 | 0.2 | 19.4 | 86.7 |
| Reading (mean) | 53.0 | 0.2 | 23.6 | 78.8 |
| Science Pipeline |  |  |  |  |
| Chemistry I or Physics I and Below | 59.7% | 1.0% | 0.0 | 100.0 |
| Chemistry I and Physics I | 19.9% | 0.9% | 0.0 | 100.0 |
| Chemistry II and Physics II | 20.5% | 0.9% | 0.0 | 100.0 |
| High School GPA (mean) | 2.8 | 0.0 | 0.0 | 4.0 |
| Mathematics Value (mean) | 2.5 | 0.0 | 1.0 | 4.0 |
| Mathematics Growth Mindset (mean) | 3.0 | 0.7 | 1.0 | 4.0 |
| *High School Characteristics* |  |  |  |  |
| Free and Reduced-Price Lunch |  |  |  |  |
| 0–5% | 21.1% | 1.5% | 0.0 | 100.0 |
| 6–20% | 25.4% | 1.6% | 0.0 | 100.0 |
| 21–50% | 37.1% | 1.7% | 0.0 | 100.0 |
| 50–100% | 16.4% | 1.2% | 0.0 | 100.0 |
| Region |  |  |  |  |
| Northeast | 19.9% | 0.9% | 0.0 | 100.0 |
| Midwest | 24.7% | 0.8% | 0.0 | 100.0 |
| South | 33.7% | 0.9% | 0.0 | 100.0 |
| West | 21.7% | 0.9% | 0.0 | 100.0 |
| Urbanicity |  |  |  |  |
| Urban | 31.0% | 0.9% | 0.0 | 100.0 |
| Suburban | 50.4% | 1.0% | 0.0 | 100.0 |
| Rural | 18.7% | 0.8% | 0.0 | 100.0 |
| *College Experiences and First Post-Secondary Institutional Characteristics* |  |  |  |  |
| Research with Faculty Outside of Class | 12.5% | 0.5% | 0.0 | 100.0 |
| Public Institution | 76.6% | 0.7% | 0.0 | 100.0 |
| Type and Selectivity |  |  |  |  |
| 2-year or Less Institution | 38.0% | 1.0% | 0.0 | 100.0 |
| 4-year Institution, Inclusive | 16.7% | 0.7% | 0.0 | 100.0 |
| 4-year Institution, Moderately Selective | 25.0% | 0.7% | 0.0 | 100.0 |
| 4-year Institution, Highly Selective | 20.3% | 0.8% | 0.0 | 100.0 |

Note: *n* = 11,535 respondents from the National Center for Education Statistics' (NCES) Education Longitudinal Study 2002/2012 restricted data. Restricted-use NCES data required rounding these descriptive results to the nearest tenth.

**Table A3.** Sample Descriptive Statistics on Dependent Variables by Gender.

|  | Men | Women | Min | Max |
|---|---|---|---|---|
| *Declared Major* | | | | |
| Undecided | 29.0% | 22.8% | 0.0 | 100.0 |
|  | (1.6%) | (1.0%) | | |
| Non-STEM | 39.5% | 45.3% | 0.0 | 100.0 |
|  | (1.4%) | (1.0%) | | |
| PEMC | 14.4% | 3.7% | 0.0 | 100.0 |
|  | (0.9%) | (0.4%) | | |
| Biological Sciences | 4.0% | 4.2% | 0.0 | 100.0 |
|  | (0.5%) | (0.4%) | | |
| Health Sciences | 3.3% | 12.7% | 0.0 | 100.0 |
|  | (0.5%) | (0.7%) | | |
| Social/Behavioral and Other Sciences | 9.7% | 11.2% | 0.0 | 100.0 |
|  | (0.7%) | (0.7%) | | |
| *Degree Major* | | | | |
| Non-STEM | 63.8% | 62.9% | 0.0 | 100.0 |
|  | (1.5%) | (1.2%) | | |
| PEMC | 13.6% | 3.6% | 0.0 | 100.0 |
|  | (0.9%) | (0.5%) | | |
| Biological Sciences | 5.5% | 4.7% | 0.0 | 100.0 |
|  | (0.6%) | (0.5%) | | |
| Health Sciences | 2.6% | 10.4% | 0.0 | 100.0 |
|  | (0.5%) | (0.7%) | | |
| Social/Behavioral and Other Sciences | 14.5% | 18.5% | 0.0 | 100.0 |
|  | (1.1%) | (1.0%) | | |

Note: *n* = 11,535 respondents from the National Center for Education Statistics' (NCES) Education Longitudinal Study 2002/2012 restricted data. Restricted-use NCES data required rounding these descriptive results to the nearest tenth. Bracketed numbers represent the standard deviation.

**Table A4.** Sample Descriptive Statistics on Dependent Variables by Race/Ethnicity.

|  | White | Asian/Pacific Islander | Black | Latino | Other | Min | Max |
|---|---|---|---|---|---|---|---|
| *Declared Major* | | | | | | | |
| Undecided | 24.2% | 27.8% | 24.5% | 33.2% | 27.3% | 0.00 | 100.00 |
|  | (1.2%) | (2.0%) | (2.0%) | (2.4%) | (4.0%) | | |
| Non-STEM | 44.4% | 31.3% | 41.7% | 38.9% | 42.2% | 0.00 | 100.00 |
|  | (1.3%) | (2.1%) | (2.2%) | (2.2%) | (4.2%) | | |
| PEMC | 8.6% | 12.7% | 11.2% | 6.5% | 8.9% | 0.00 | 100.00 |
|  | (0.5%) | (1.5%) | (1.3%) | (1.1%) | (1.9%) | | |
| Biological Sciences | 4.2% | 8.3% | 3.2% | 3.2% | 3.1% | 0.00 | 100.00 |
|  | (0.4%) | (1.1%) | (0.7%) | (0.7%) | (1.3%) | | |
| Health Sciences | 7.5% | 9.2% | 11.1% | 8.1% | 7.9% | 0.00 | 100.00 |
|  | (0.5%) | (1.3%) | (1.4%) | (1.3%) | (2.2%) | | |
| Social/Behavioral and Other Sciences | 11.0% | 10.6% | 8.3% | 10.1% | 10.6% | 0.00 | 100.00 |
|  | (0.6%) | (1.3%) | (1.1%) | (1.5%) | (2.5%) | | |
| *Degree Major* | | | | | | | |
| Non-STEM | 64.7% | 50.5% | 63.8% | 61.8% | 61.0% | 0.00 | 100.00 |
|  | (1.2%) | (2.2%) | (2.9%) | (3.0%) | (3.6%) | | |
| PEMC | 8.7% | 12.7% | 8.1% | 6.7% | 6.9% | 0.00 | 100.00 |
|  | (0.7%) | (1.4%) | (1.5%) | (1.1%) | (1.7%) | | |
| Biological Sciences | 4.9% | 11.4% | 3.8% | 4.3% | 5.8% | 0.00 | 100.00 |
|  | (0.6%) | (1.3%) | (0.9%) | (1.1%) | (1.9%) | | |
| Health Sciences | 6.5% | 7.3% | 7.8% | 6.0% | 6.6% | 0.00 | 100.00 |
|  | (0.5%) | (1.0%) | (1.5%) | (1.4%) | (2.1%) | | |
| Social/Behavioral and Other Sciences | 15.2% | 18.1% | 16.5% | 21.2% | 19.7% | 0.00 | 100.00 |
|  | (0.9%) | (1.6%) | (2.0%) | (2.4%) | (3.6%) | | |

Note: *n* = 11,535 respondents from the National Center for Education Statistics' (NCES) Education Longitudinal Study 2002/2012 restricted data. Restricted-use NCES data required rounding these descriptive results to the nearest tenth. Bracketed numbers represent the standard deviation.

**Table A5.** PEMC Outcomes by Sex, Race/Ethnicity, and Difficulty Orientations.

| | Declared | | Degree Field | |
|---|---|---|---|---|
| | **RRR** | **SE** | **RRR** | **SE** |
| *Demographic Characteristics* | | | | |
| Sex (Reference = Male) | | | | |
| Female | 0.24 *** | 0.05 | 0.27 *** | 0.06 |
| Race/Ethnicity (Reference = White) | | | | |
| Asian/Pacific Islander | 1.23 | 0.38 | 1.03 | 0.31 |
| Black | 2.23 ** | 0.58 | 1.51 | 0.43 |
| Latino | 1.23 | 0.41 | 1.06 | 0.37 |
| Other | 1.52 | 0.77 | 1.01 | 0.46 |
| *Difficulty Orientations* | | | | |
| General Academic Scale | 0.94 | 0.14 | 0.98 | 0.15 |
| Verbal Scale | 0.74 | 0.09 | 0.69 ** | 0.08 |
| Mathematics Scale | 1.72 ** | 0.26 | 1.46 | 0.22 |
| *Demographic Characteristics Interactions* | | | | |
| Female × Asian/Pacific Islander | 1.13 | 0.45 | 1.28 | 0.53 |
| Female × Black | 1.15 | 0.46 | 0.99 | 0.51 |
| Female × Latino | 1.26 | 0.66 | 1.41 | 0.72 |
| Female × Other | 0.82 | 0.69 | 1.21 | 1.04 |
| *Mathematics Difficulty Orientation Interactions* | | | | |
| Female × Mathematics Scale | 0.88 | 0.19 | 0.84 | 0.19 |
| Asian/Pacific Islander × Mathematics Scale | 0.81 | 0.28 | 0.88 | 0.27 |
| Black × Mathematics Scale | 0.80 | 0.24 | 0.99 | 0.26 |
| Latino × Mathematics Scale | 0.71 | 0.23 | 1.02 | 0.39 |
| Other × Mathematics Scale | 0.61 | 0.27 | 0.53 | 0.26 |
| *Demographics and Mathematics Difficulty Orientation Interactions* | | | | |
| Female × Asian/Pacific Islander × Mathematics Scale | 1.35 | 0.61 | 1.34 | 0.58 |
| Female × Black × Mathematics Scale | 1.15 | 0.56 | 1.58 | 0.83 |
| Female × Latino × Mathematics Scale | 1.29 | 0.87 | 0.97 | 0.58 |
| Female × Other × Mathematics Scale | 0.77 | 0.84 | 2.35 | 2.38 |
| Constant | 0.00 *** | 0.00 | 0.00 *** | 0.00 |
| *f*-statistic | 5.09 *** | | 3.77 *** | |
| Observations | 11,535 | | 11,535 | |

Note. $n$ = 11,535 respondents from the National Center for Education Statistics' Education Longitudinal Study 2002/2012 restricted data. Parent education, family income, 10th grade standardized test scores, science course taking, high school GPA, mathematics value, mathematics growth mindset, percentage free and reduced-price lunch, high school region, high school urbanicity, participation in undergraduate research, institutional control, and college selectivity was included in the model, but not shown for space. Full table is available upon request. * $p < 0.05$, ** $p < 0.01$, *** $p < 0.001$.

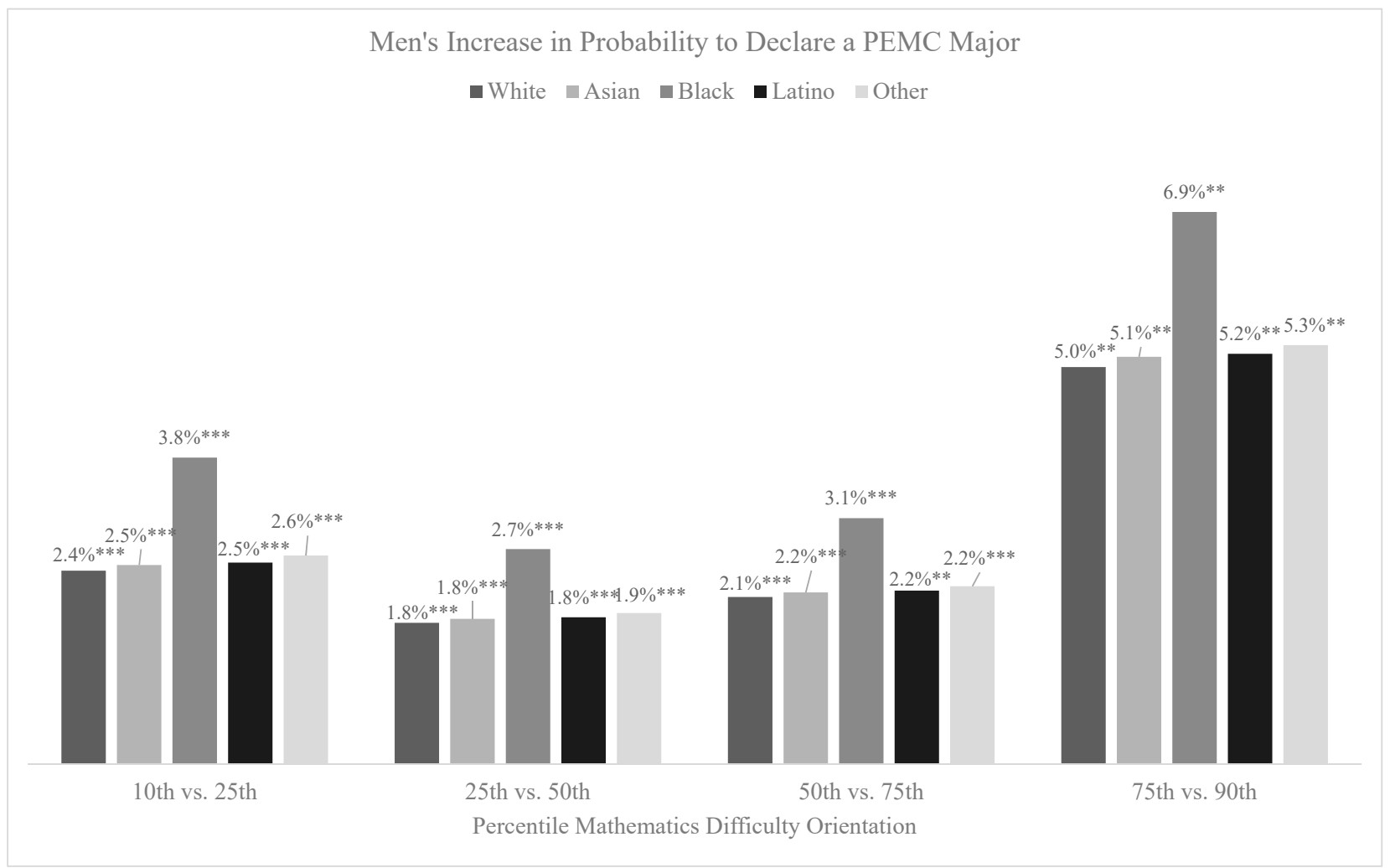

**Figure A1.** Men's Increase in Probability to Declare a PEMC Major Given Mathematics Difficulty Orientation.

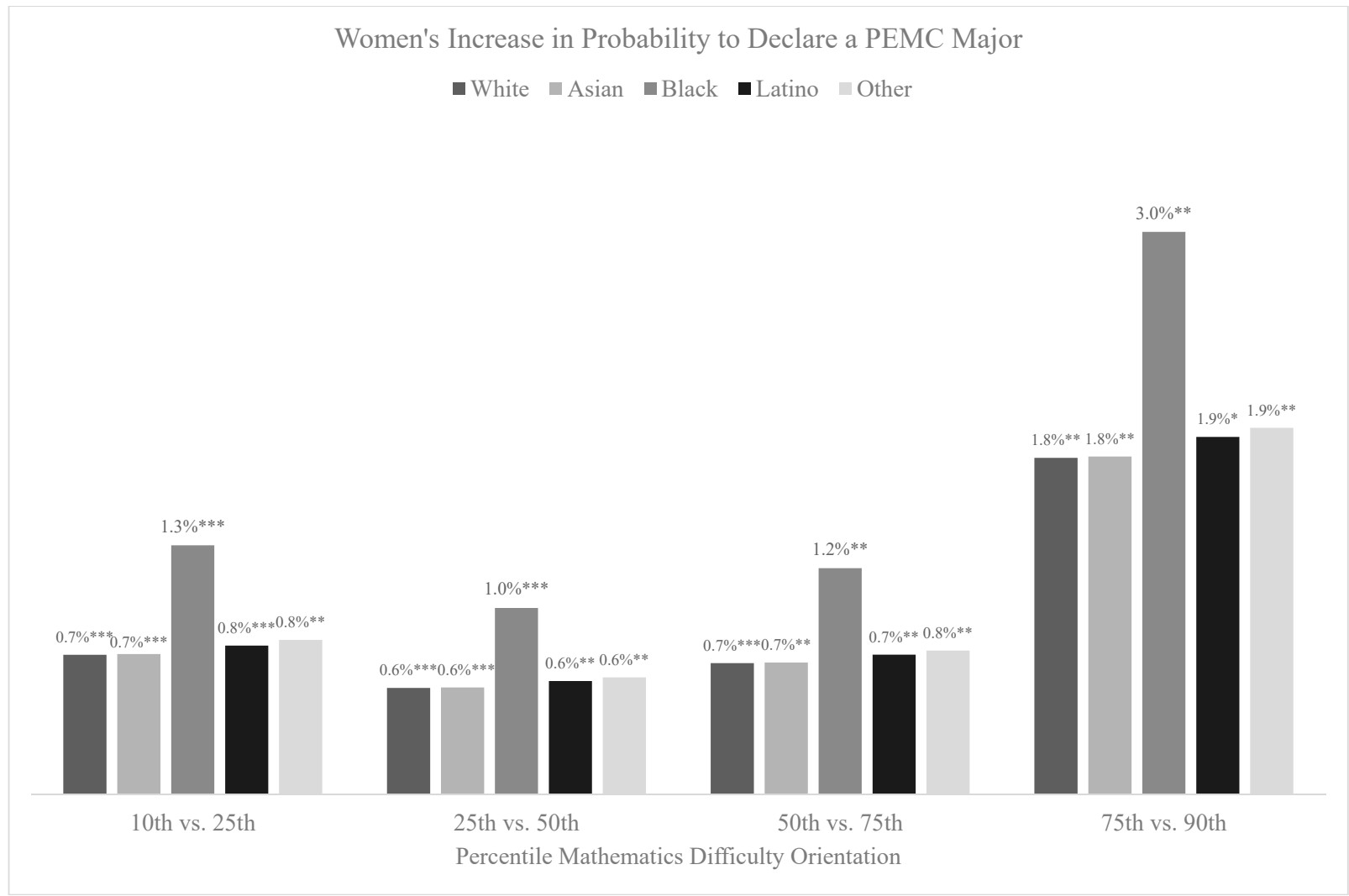

**Figure A2.** Women's Increase in Probability to Declare a PEMC Major Given Mathematics Difficulty Orientation.

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
