# Peer review of "Difficulty Orientations, Gender, and Race/Ethnicity: An Intersectional Analysis of Pathways to STEM Degrees"

_socsci, doi:10.3390/socsci8020043_

Round 1
Reviewer 1 Report
This study explores the relationship between mathematics ability beliefs and STEM degrees, with particular attention to gender and race-ethnicity. The topic is important and the study is capably conducted. The study would be enhanced with attention to the following considerations.
There is some theoretical diffuseness exhibited in the manuscript. Learning theories are mentioned as is coping theory and flow theory. These are useful and relevant, but can they be grouped under a broad theoretical umbrella to clarify the relationship between them? Highlighting the power of perception is important, so if there is a broad paradigm under which all of these theories can be grouped, that would clarify the argumentation in that part of the manuscript. In one respect, all of these approaches underscore the power of cultural perceptions as opposed to structural factors. So, perhaps this study can be seen as defining and applying a *cultural model* that is played off against the extensive examination of structural factors. I also wonder if it is worth citing W.I. Thomas's statement which argues that situations defined as real are real in their consequences (paraphrased here).
I am glad that the authors controlled for parental education. However, is there a variable on parental occupation, particularly by parents' gender? If so, please control for that as well. See the previously published study in Social Sciences by Bowden et al. that tests a social reproduction model using this factor.
I would encourage the authors to end their review of the literature with hypotheses that state anticipated relationships. The RQs articulated in the Analysis section are good, but could be well complemented by hypotheses conveyed earlier than that.
The arguments about gender and race-ethnicity variations could be clarified in various spots. The use of the word "and" suggests an intersectionality approach. If the authors were testing for respective influences by gender and thereafter by race-ethnicity, they could use more precise language (e.g., we examine variations by gender and by race-ethnicity, respectively). However, as I see from from the figures, they are using an intersectional approach in which gender is "raced" and race-ethnicity is "gendered." Given this, some mention of intersectionality theory or statistical applications of this theory seems warranted. Perceptions are not merely shaped independently by gender and by race-ethnicity, but by the intersection of these social categories.
This is a solid study with some room for improvement.
Author Response
Thank you for your thoughtful and constructive review.
Point 1: Theoretical diffuseness. Learning theories are mentioned as is coping theory and flow theory. These are useful and relevant, but can they be grouped under a broad theoretical umbrella to clarify the relationship between them? Highlighting the power of perception is important, so if there is a broad paradigm under which all of these theories can be grouped, that would clarify the argumentation in that part of the manuscript. In one respect, all of these approaches underscore the power of cultural perceptions as opposed to structural factors. So, perhaps this study can be seen as defining and applying a *cultural model* that is played off against the extensive examination of structural factors. I also wonder if it is worth citing W.I. Thomas's statement which argues that situations defined as real are real in their consequences.
Response 1:
1. We appreciate this constructive and thoughtful critique.
2. We sharpened and honed the language in the introduction to enhance the theoretical sharpness.
3. On page 1, we took your suggestion about the Thomas Theorem, and cited Robert Merton’s Social Forces article on the Thomas effect, giving a nod to W.I and D.S. Thomas’s important sociological work (1928) on the real-world consequences of children’s beliefs as well as R.K. Merton’s work on self-fulfilling prophecies.
4. Additional theoretical clarity is offered in the paragraph on the top of page 7.
Point 2: Parental education variable. Is there a variable on parental occupation, particularly by parents' gender? If so, please control for that as well (See Bowden et al., 2018).
Response 2: Thank you for this suggestion. The Bowden et al., 2018 paper using ECLS data does find compelling associations between parental occupation and girls’ academic advantages in elementary and middle school. The evidence for effects from upper secondary school to postsecondary is more mixed (Cheng, Kopoptic, & Zamarro, 2017; Engberg & Wolniak, 2013). The newer NCES cohort following ELS – the dataset we use in the paper – has a stronger STEM focus in its secondary school and background variables. The parental occupation variables (by parent/guardian) are limited in its utility in the data we have available.
One of the authors has done extensive manipulation of these variables after eventually ruling it out to have a stronger and more parsimonious set of predictors over the multiple waves and ten-year time span covered in the study. The categorical variable designates rank rather than discipline or domain; parental STEM occupations are not split out and not specifically designated as distinct from other professional classes. We agree this is an intriguing predictor to pursue with other data.
Although we do not have this variable available, we do have other family background variables of importance including parental social class indicators like parental education and family income, which themselves are associated with parental occupation.
Related to issues of pre-college opportunity and access to information through their secondary schooling, we also have indicators of school SES which are also highly associated with school academic and social resources, and have detailed measures on science course taking in addition to test scores.
Point 3: Hypotheses. I would encourage the authors to end their review of the literature with hypotheses that state anticipated relationships.
Response 3:Thank you, we have added hypotheses immediately after the research questions in the introduction.
Point 4: Clarification of gender and race-ethnicity variation. The use of the word "and" suggests an intersectionality approach. If the authors were testing for respective influences by gender and thereafter by race-ethnicity, they could use more precise language (e.g., we examine variations by gender and by race-ethnicity, respectively). However, as I see from the figures, they are using an intersectional approach in which gender is "raced" and race-ethnicity is "gendered." Given this, some mention of intersectionality theory or statistical applications of this theory seems warranted. Perceptions are not merely shaped independently by gender and by race-ethnicity, but by the intersection of these social categories.
Response 4:
Our thanks for illuminating this issue. This paper emerged from a larger mixed methods study that examined the gendered and raced experiences of diverse students, particularly in reference to their difficulty perceptions in STEM. The approach here reflects our best efforts to tell both the raced and gendered stories of participants via the available national statistics. However, we recognize that our discussion on intersectionality theory merited deeper attention. We have added to our framing of intersectionality theory in the literature review.
Reviewer 2 Report
Overall, the manuscript provides interesting findings in the field of STEM participation. The longitudinal data and the differentiated view on gender and race/ethnical background are convincing. Still, in my opinion the manuscript needs several changes in the introduction of the investigated constructs, in the description of the hypotheses, in the presentation of the results, and in the discussion of the findings.
Major points:
Introduction
Throughout the abstract and the manuscript authors use the expressions “ability beliefs”, “difficulty perception” and “difficulty orientation” (see p.3). In my impression, authors use these expressions synonymously. This is misleading and even incorrect. Whereas ability beliefs refer to the perception of ones owns characteristics, perceived difficulty refers to perceived characteristics of tasks or domains. It is important to describe the theoretical construct more precisely.
On pp.5-6 the authors describe the perceptions of talent, challenge and difficulty as the main variables of interest in this study. However, later in the manuscript they call the measure of main interest “difficulty orientations”. It is not clear to me, why the authors chose this expression instead of using the well-known expression “ability beliefs” or even “self-concept”. The description of self-concept and self-efficacy is short and not exhaustive and I would recommend giving a more detailed overview about both concepts (see e.g., Bong & Skaalvik, 2003). The sentence “Foundational self-concept research shows that students tend to believe that they have ability in either mathematics or verbal domains, …” on p.6 needs more explication. I assume this refers to the Internal/External frame of Reference Model by Marsh (1986). This should be mentioned/described and cited (see e.g., Möller, Pohlmann, Köller, & Marsh, 2009; Möller, Marsh, 2013). Furthermore, to point out domain-specifity of academic self-concepts, authors should include the corresponding literature (e.g., Marsh, & Shavelson 1985)
Method
The description of the independent variables leaves several questions unanswered (p.9). How many items were used to asses “difficulty orientation”. Are the items in Table A1 example items or are all items included in the table? Which origin do these items have? Are they part of a well know scale? Did participants answer the items on a Likert-scale? What about the reliability of the measures? Why do the authors call the construct “difficulty orientation”? This construct was not defined or cited in the introduction. These points should be clarified in the revision.
What are the hypotheses of the authors? After naming the research questions authors should formulate specific and theory driven hypotheses about gender differences, relations between “difficulty orientations” and PEMC degrees, and interactions between gender and race/ethnicity.
The authors describe that variables were progressively introduced in a series of multinominal logistic regression models. This procedure can be problematic for the generic logistic regression analysis because the dependent variable is rescaled in every model, and hence, the coefficients of nested models are not comparable. Karlson et al. (2012) give a good overview about this methodological challenge. Using average marginal effects (AMEs) or using the KHB method by Karlson, Holm and Breen (2012) would be both appropriate approaches to deal with this methodological issue. I am not an expert with STATA commands, so I am not sure, if the “post-estimation predicted probabilities” are based on estimated AMEs. The description of the predicted probabilities indicates this. However, the cited literature (Klein 2016) is not included in the references of the manuscript. Please include the reference and a more detailed description of the predicted probabilities.
By using the KHB method, residuals from all the predictors in the full model are included in each of the nested models. Thus, the variance of all models is fixed, and the coefficients in the nested models are comparable (Karlson et al., 2012; Kohler, Karlson, & Holm, 2011).This method is implemented in STATA and could be applied by the authors. However, the KHB method was not designed for imputed data and hence some difficulties could occur by using the KHB method in this study. If not applicable, I would recommend including and interpreting only the full models in the results.
Results
As mentioned earlier, comparison of effects in nested models in logistic regression models should be treated with caution. Why not including only the full model (Base + All.D.O.) and interpret only these results? I would seriously recommend excluding the comparison of PPs of the progressive logistic regression models (see pp.16-17). Differences in the PPs are small, authors do not provide information about significance of these differences and differences could be a result of the rescaling of the dependent variable in the models. However, if authors decide including AMEs and/or using the KHB method, results of nested models could be compared.
I was wondering why the results of multinominal logistic regression models with interaction terms are not included in a table. In my impression these are interesting findings and central results concerning research question 3 and should therefore be explained more explicitly.
Minor points:
Abstract:
The last sentence in the abstract indicates a significant interaction effect between gender and race/ethnic identities and the relation between ability beliefs and STEM outcomes. However, results showed significant main effects of gender and race on ability beliefs and STEM outcomes, but no statistically significant interaction effects (i.e., different intercepts, no different slopes). This should be described more precisely in the abstract
Introduction
On p. 2 the authors point out that “women and racial/ ethnic minority groups may be negatively affected by cultural stereotypes …”. I would recommend describing this relation in more detail: Do stereotypes negatively affect their learning outcomes in mathematical domains? Or are stereotypes related to ability beliefs in STEM domains? This would help the reader to follow this part of the introduction.
I could not follow the argument in the first sentence of the second paragraph on p.2. Indeed, ability beliefs are important for course enrollment in secondary school. And course enrollment in secondary school is related to college major choices. If authors refer to this relation, they should describe it in more detail and cite the corresponding literature. Otherwise, I would exclude this sentence from the manuscript, as course enrollment in high school is not the focus of the present study.
On p.5 authors mention the relation between institutional characteristics and the persistence of women and underrepresented minority groups in STEM. I would recommend including a short description about how these characteristics are related to the persistence in STEM. Otherwise, this information is little helpful for the reader. Similar, the authors mention that college student experiences have been investigated in the context of post-secondary STEM outcomes without describing the results. These should be included in the manuscript.
The last sentence of the second paragraph on p.5 is not fully clear to me.
The concept of “growth mindset” is mentioned on p.6 without further explanation. A definition and description should be included in the manuscript.
Method
In the footnote on p.8 it would be easier for the reader to know the missing rate in %.
The first sentence of the paragraph “covariates” gives the impression that the relation of postsecondary experiences and “difficulty orientations” were examined in the study. To my understanding this was not the case. Therefore, I would change this sentence.
If included in the study, the constructs and measures of valuing mathematics and mathematics growth mindset should be explained in one or two sentences.
Results
It would be helpful to know the range of the scale on which participants answered the items of “difficulty orientation”. Table A1 does not provide this information.
In my opinion the sentence on p.14 “We found clear differences in mean difficulty orientations between men and women, but more variable differences between White and non-White students.” Is a bit misleading. I would have expected larger differences in the “difficulty orientation” between White and non-White students. However the difference of .4 between boys and girls “difficulty orientation” in math is more pronounced than the differences in “difficulty orientation” in all domains between White and non-White students. For better understanding I would change the sentence.
I would recommend including the tables displaying the results for the other but PEMC domains in the supplemental material.
I would recommend including information about the residual variances or R² of the models.
The Figures 1, 2, A1 and A2 are very insightful. I would recommend linking the figures more precisely to the corresponding parts in the text on p.18. So far it is a bit confusing which information can be found in the Appendix and in the manuscript.
On p.18 in the paragraph “Race-ethnicity” authors describe differences in the probability of earning a PEMC degree by race/ethnicity (“With respect to degrees earned, other race/ethnicity students had the lowest probability of earning PEMC degrees, even as they were modestly higher than all other non-Black students with respect to declaring PEMC majors”). Were these differences statistically significant?
On p.18, I assume that authors refer to the Figures A1 and A2 in the appendix, rather than to Figures 1 and 2?
Discussion
The findings of the study are interesting and insightful. I would appreciate a more fine-grained discussion of the results. How do the authors explain the results of black men and women having a higher probability of enrolling and graduation in PEMC studies, compared to other groups, when achievement, “difficulty orientation” and several other covariates are controlled? Especially, when statistics show that Black students still are a minority among students in the PEMC field (see also p.3). How do the authors interpret the finding of no statistically significant interaction effects between gender and race/ethnicity?
In the “Implication” (p.21) authors again use the expression ability beliefs instead of “difficulty orientation”. It is important to use a consistent language throughout the whole manuscript and to introduce and explain the investigated construct precisely in the introduction.
In the limitation authors mention that the timespan of 10 years could be too long to capture interaction effects of gender and race/ethnicity. I do not completely understand this argument and would appreciate a broader explanation. Authors also mention small sample sizes of intersectional groups to possibly be problematic for the analyses. It would be helpful to provide information about the sample sizes in the intersectional groups. So far this information is not given in the manuscript.
Author Response
Thank you for this thoughtful and constructive review.
Point 1: Word confusion.
Confusion among the words “ability beliefs”, “difficulty perception” and “difficulty orientation” (see p.3). In my impression, authors use these expressions synonymously. This is misleading and even incorrect. Whereas ability beliefs refer to the perception of ones owns characteristics, perceived difficulty refers to perceived characteristics of tasks or domains. It is important to describe the theoretical construct more precisely.
Confusion about the word “difficulty orientations”. On pp.5-6 the authors describe the perceptions of talent, challenge and difficulty as the main variables of interest in this study. However, later in the manuscript they call the measure of main interest “difficulty orientations”. It is not clear to me, why the authors chose this expression instead of using the well-known expression “ability beliefs” or even “self-concept”.
Response 1: We appreciate this comment. We have made edits throughout the introduction and literature review to further drive home our dual interest in perceived ability and difficulty through what we are calling “difficulty orientations” in this paper. Particularly, we recognize the confusion of using so many terms. We have made edits in an effort to streamline these ideas, pulling forward the main constructs of perceived ability and difficulty while still recognizing the central theories from which this study is built. We have also added a few more citations in the literature review to describe our interest in difficulty perceptions to help balance our discussion of ability beliefs, as we believe that some of the confusion was due to an imbalance in these citations. We believe that, given these edits, our choice to focus the study on “difficulty orientations” rather than the more general “ability beliefs” or “self-concepts” is clear to the readers.
Point 2: Concepts. The description of self-concept and self-efficacy is short and not exhaustive and I would recommend giving a more detailed overview about both concepts (see e.g., Bong & Skaalvik, 2003). The sentence “Foundational self-concept research shows that students tend to believe that they have ability in either mathematics or verbal domains, …” on p.6 needs more explication. I assume this refers to the Internal/External frame of Reference Model by Marsh (1986). This should be mentioned/described and cited (see e.g., Möller, Pohlmann, Köller, & Marsh, 2009; Möller, Marsh, 2013). Furthermore, to point out domain-specifity of academic self-concepts, authors should include the corresponding literature (e.g., Marsh, & Shavelson 1985)
Response 2: Thank you for the rigorous engagement of these foundational theories. We appreciate the suggestion to broaden discussion of both self-efficacy and self-concept and have made edits to better address these theories. We also appreciate the additional citations for the I/E model, which we have integrated in the literature review.
Point 3: Independent variables (difficulty orientation). How many items were used to asses “difficulty orientation”? Are the items in Table A1 example items or are all items included in the table? Which origin do these items have? Are they part of a well know scale? Did participants answer the items on a Likert-scale? What about the reliability of the measures? Why do the authors call the construct “difficulty orientation”? This construct was not defined or cited in the introduction. These points should be clarified in the revision.
Response 3: Thank you for this comment. We agree that readers deserve more detail on this topic and have integrated edits on p. 8, 9-10, and Table A1.
Point 4: Hypotheses. After naming the research questions authors should formulate specific and theory driven hypotheses about gender differences, relations between “difficulty orientations” and PEMC degrees, and interactions between gender and race/ethnicity
Response 4: Thank you, we have added hypotheses in the space suggested.
Point 5: Approach. The authors describe that variables were progressively introduced in a series of multinomial logistic regression models. This procedure can be problematic for the generic logistic regression analysis because the dependent variable is rescaled in every model, and hence, the coefficients of nested models are not comparable. Karlson et al. (2012) give a good overview about this methodological challenge. Using average marginal effects (AMEs) or using the KHB method by Karlson, Holm and Breen (2012) would be both appropriate approaches to deal with this methodological issue. I am not an expert with STATA commands, so I am not sure, if the “post-estimation predicted probabilities” are based on estimated AMEs. The description of the predicted probabilities indicates this. However, the cited literature (Klein 2016) is not included in the references of the manuscript. Please include the reference and a more detailed description of the predicted probabilities.
By using the KHB method, residuals from all the predictors in the full model are included in each of the nested models. Thus, the variance of all models is fixed, and the coefficients in the nested models are comparable (Karlson et al., 2012; Kohler, Karlson, & Holm, 2011). This method is implemented in STATA and could be applied by the authors. However, the KHB method was not designed for imputed data and hence some difficulties could occur by using the KHB method in this study. If not applicable, I would recommend including and interpreting only the full models in the results.
Response 5: Thank you for this comment and noting that the Klein citation was missing the author, and therefore was difficult to identify in the references. We have resolved this Endnote reference glitch, to allow readers to investigate the mimrgns command further. Based on documentation, the mimrgns command builds on and uses the built-in Stata margins command, which as you pointed out, is derived from the estimated average marginal effects (AMEs). This is no different in the mimrgns command. The additional benefit to using the mimrgns command is that it then applies Rubin’s rules to accurately reflect variation across the multiply-imputed datasets. Given that the KHB method is not available using multiply-imputed data and that the mimrgns command does use AMEs, we hope that our continued use of predicted probabilities satisfies your nuanced and appreciated methodological suggestions. We further note that Karlson, Holm and Breen (2012) state that comparing coefficients of nested models underestimates potential role of a z variable (in this case, difficulty orientations). Therefore, we believe that this method is actually the more conservative approach.
Point 6: Results. Comparison of effects in nested models in logistic regression models should be treated with caution. Why not including only the full model (Base + All.D.O.) and interpret only these results? I would seriously recommend excluding the comparison of PPs of the progressive logistic regression models (see pp.16-17). Differences in the PPs are small, authors do not provide information about significance of these differences and differences could be a result of the rescaling of the dependent variable in the models. However, if authors decide including AMEs and/or using the KHB method, results of nested models could be compared.
Response 6: Thank you for the constructive suggestion. We would like to clarify that we do show the statistical significance of these differences in Tables 3 and 4, and there is some variation between each of the models as well as between the progressive and full models. We certainly want to present the strongest models and paper we can. We do believe however that there is added value from including the full set of models to demonstrate the shifts in significance among the predictors, and especially the importance of only domain-specific predictors in our model, including quite notably verbal difficulty orientation. As described above, estimates from our models are consistent with the procedure which derives AMEs and is therefore according to the reviewer in the point above, appropriately suited for “results of nested models” to be compared.
Point 7: Results of model with interaction terms. I was wondering why the results of multinominal logistic regression models with interaction terms are not included in a table. In my impression these are interesting findings and central results concerning research question 3 and should therefore be explained more explicitly.
Response 7: We address the lack of tables with interaction terms on p. 18-19, “Intersectional analyses” under “RQ3. Do the relationships between difficulty orientations and PEMC outcomes vary by gender and race/ethnicity?” The coefficients for the interaction terms were not significant, therefore for space and brevity, we chose not to show these tables. We explain to readers that the lack of significance indicates that there are no true slope differences between gender and race/ethnicity groups—probability to participate in PEMC given one’s mathematics difficulty orientation generally moves in the same direction for all groups. However, there were nuanced differences in predicted probabilities by race/ethnicity group that we described in the results.
Minor points:
Abstract:
The last sentence in the abstract indicates a significant interaction effect between gender and race/ethnic identities and the relation between ability beliefs and STEM outcomes. However, results showed significant main effects of gender and race on ability beliefs and STEM outcomes, but no statistically significant interaction effects (i.e., different intercepts, no different slopes). This should be described more precisely in the abstract.
Response 8: We appreciate your suggestion to be more precise and have edited the sentence with your critique in mind.
Introduction
On p. 2 the authors point out that “women and racial/ ethnic minority groups may be negatively affected by cultural stereotypes …”. I would recommend describing this relation in more detail: Do stereotypes negatively affect their learning outcomes in mathematical domains? Or are stereotypes related to ability beliefs in STEM domains? This would help the reader to follow this part of the introduction.
Response 9: Thank you. We added further clarification here.
I could not follow the argument in the first sentence of the second paragraph on p.2. Indeed, ability beliefs are important for course enrollment in secondary school. And course enrollment in secondary school is related to college major choices. If authors refer to this relation, they should describe it in more detail and cite the corresponding literature. Otherwise, I would exclude this sentence from the manuscript, as course enrollment in high school is not the focus of the present study.
Response 10: We understand this point and for brevity have cut the sentence.
On p.5 authors mention the relation between institutional characteristics and the persistence of women and underrepresented minority groups in STEM. I would recommend including a short description about how these characteristics are related to the persistence in STEM. Otherwise, this information is little helpful for the reader. Similar, the authors mention that college student experiences have been investigated in the context of post-secondary STEM outcomes without describing the results. These should be included in the manuscript.
The last sentence of the second paragraph on p.5 is not fully clear to me.
Response 11: Edits have been integrated in this paragraph to provide more clarity.
The concept of “growth mindset” is mentioned on p.6 without further explanation. A definition and description should be included in the manuscript.
A brief description has been included in what is now the top of page 8.
Method
In the footnote on p.8 it would be easier for the reader to know the missing rate in %.
Response 12: The proportion has been calculated for the reader and included in the footnote.
The first sentence of the paragraph “covariates” gives the impression that the relation of postsecondary experiences and “difficulty orientations” were examined in the study. To my understanding this was not the case. Therefore, I would change this sentence.
Response 13: Thank you for this observation. We have edited the sentence accordingly.
If included in the study, the constructs and measures of valuing mathematics and mathematics growth mindset should be explained in one or two sentences.
Response 14: We added an explanation of “valuing mathematics” and kept the explanation of growth mindset, both included as footnotes. Since these measures were controls and not constructs of interest, we believe that these explanations are sufficient for the readers.
Results
It would be helpful to know the range of the scale on which participants answered the items of “difficulty orientation”. Table A1 does not provide this information.
Response 15: The table has been edited to include this information (page 32; article Appendix).
In my opinion the sentence on p.14 “We found clear differences in mean difficulty orientations between men and women, but more variable differences between White and non-White students.” Is a bit misleading. I would have expected larger differences in the “difficulty orientation” between White and non-White students. However the difference of .4 between boys and girls “difficulty orientation” in math is more pronounced than the differences in “difficulty orientation” in all domains between White and non-White students. For better understanding I would change the sentence.
Response 16: We edited the sentence to emphasize the clearer difference by gender vs. race/ethnicity.
I would recommend including the tables displaying the results for the other but PEMC domains in the supplemental material.
Response 17: We appreciate this recommendation. Although we accommodated the other requests here and with the other reviewer in the 10-day revision window we were given, upon review, we find it too cumbersome for the reader and may distract from our key presentations of data and models herein, as there are already four tables and two figures (with two graphs per figure) in the main manuscript, plus an additional four tables and two figures in the appendix/supplement. Adding additional tables has the potential risk of distracting the reader from the central arguments in an already data-intensive, theory-rich manuscript with an interest in intersectionality and discipline-specificity within STEM. In addition, the manuscript is already longer post-revisions in response to requested additions. We do contrast PEMC and other STEM field clusters in prior papers, but those do not do not look with intention at racial/ethnic and gender differences within these fields. To emphasize this intersectional focus, we narrowed our focus to emphasize patterns within these key mathematically-intensive STEM fields. While the authors appreciate the interest in even more data to peruse, we respectfully decline this request.
I would recommend including information about the residual variances or R² of the models.
Response 18: Thank you for this comment. We used the most recent and most advanced version of Stata’s analytic software, but still Stata’s mlogit command with multiply-imputed data does not offer an R²; rather, it produces f-statistic to designate model fit.
The Figures 1, 2, A1 and A2 are very insightful. I would recommend linking the figures more precisely to the corresponding parts in the text on p.18. So far it is a bit confusing which information can be found in the Appendix and in the manuscript.
On p.18 in the paragraph “Race-ethnicity” authors describe differences in the probability of earning a PEMC degree by race/ethnicity (“With respect to degrees earned, other race/ethnicity students had the lowest probability of earning PEMC degrees, even as they were modestly higher than all other non-Black students with respect to declaring PEMC majors”). Were these differences statistically significant?
On p.18, I assume that authors refer to the Figures A1 and A2 in the appendix, rather than to Figures 1 and 2?
Response 19: Thank you for these comments, and for the compliment on our figures. We have added parenthetical references to tables and figures as appropriate throughout the findings and especially here on what is now page 19+. In addition, we have edited our discussion of race/ethnicity findings to more fully and sharply represent our findings and reduce any confusion.
Discussion
The findings of the study are interesting and insightful. I would appreciate a more fine-grained discussion of the results. How do the authors explain the results of black men and women having a higher probability of enrolling and graduation in PEMC studies, compared to other groups, when achievement, “difficulty orientation” and several other covariates are controlled? Especially, when statistics show that Black students still are a minority among students in the PEMC field (see also p.3).
Response 20: Thank you for this comment, illuminating that our original discussion of this on p. 26 (the paragraph starting, “Given Black college degree-earners’ probability of earning PEMC …”) could be more clearly addressed. We have edited the discussion to more clearly discuss the potential that structural racism is at play. This edit occurs earlier in the narrative, to further highlight this issue.
In the “Implication” (p.21) authors again use the expression ability beliefs instead of “difficulty orientation”. It is important to use a consistent language throughout the whole manuscript and to introduce and explain the investigated construct precisely in the introduction.
Response 21: We agree with this comment and thank the reviewer for pointing this inconsistency out. We have made edits accordingly and re-reviewed the manuscript as a whole for linguistic consistency.
How do the authors interpret the finding of no statistically significant interaction effects between gender and race/ethnicity?
In the limitation authors mention that the timespan of 10 years could be too long to capture interaction effects of gender and race/ethnicity. I do not completely understand this argument and would appreciate a broader explanation. Authors also mention small sample sizes of intersectional groups to possibly be problematic for the analyses. It would be helpful to provide information about the sample sizes in the intersectional groups. So far this information is not given in the manuscript.
Response 22: Thank you for these comments. We have added a paragraph right before the Implications section to dig a bit more deeply in the lack of significant results for the interaction models. In addition, we edited the language under the Limitations section to provide more clarity. We provided more specifics on the small sample size and removed the mention of the 10-year timespan for brevity and to limit potential confusion.
Round 2
Reviewer 1 Report
I commend the authors on an excellent revision. I have no further concerns. I’d suggest that the authors consider replacing the reference to ability beliefs in the title with difficulty orientations.
Author Response
Dear Reviewer,
Thank you for the encouraging comments. We concur with your suggestion and will update the title accordingly.
Thank you, the authors.
Reviewer 2 Report
The authors addressed most of the concerns in revised manuscript. The theoretical introduction is more clearly and the explanation of the statistical analysis is clearer to me after reading the response letter and the revised manuscript. Still I have some minor points that should be addressed before publication.
1. I appreciate that the authors included the hypothesis in the manuscript. I wondered why the hypotheses were included in the introduction. I recommend including the hypotheses in the section of the analysis along the corresponding research questions
(E.g., RQ1: xxx
H1: xxx)
2. The sentence “We refined the scales from the first study to this study, and use this term as a way to distinguish it from the first study” on p. 8 is not fully clear to me. I recommend to change the expression “first study” in the correct citation
3. I repeat my recommendation to include the results of the interaction effects in a table, even if they are not significant. It is an important result that main effects of gender and race/ethnicity occurred, but interaction effects were not statistically significant. Hence, differences in the intercepts occurred (even among different groups of race/ethnicity), whereas the slopes did not differ between groups. If the authors prefer not to include an additional table in the manuscript, I seriously recommend including these results in the appendix or supplemental material.
4. Thanks for the clarification of the predicted probabilities. I was wondering why PPs were not reported for the scales of “difficulty orientations”?
Author Response
The authors addressed most of the concerns in revised manuscript. The theoretical introduction is more clearly and the explanation of the statistical analysis is clearer to me after reading the response letter and the revised manuscript. Still I have some minor points that should be addressed before publication.
1. I appreciate that the authors included the hypothesis in the manuscript. I wondered why the hypotheses were included in the introduction. I recommend including the hypotheses in the section of the analysis along the corresponding research questions
(E.g., RQ1: xxx
H1: xxx)
Thank you for this suggestion. We have made this exact change in the manuscript (page 11).
2. The sentence “We refined the scales from the first study to this study, and use this term as a way to distinguish it from the first study” on p. 8 is not fully clear to me. I recommend to change the expression “first study” in the correct citation.
We appreciate this sentence may have been unclear. We honed the sentence and the one preceding it, to what we believe is a sharper and easier to follow articulation of what we did and why: “We refined the scales since the earlier study and here refer to these ability beliefs as difficulty orientations, uniquely focusing on challenging academic work, especially in mathematics” (page 8).
3. I repeat my recommendation to include the results of the interaction effects in a table, even if they are not significant. It is an important result that main effects of gender and race/ethnicity occurred, but interaction effects were not statistically significant. Hence, differences in the intercepts occurred (even among different groups of race/ethnicity), whereas the slopes did not differ between groups. If the authors prefer not to include an additional table in the manuscript, I seriously recommend including these results in the appendix or supplemental material.
Thank you for this comment. We appreciate that readers may still want to see some data (and lack of significance) for the interaction terms. As recommended, we have now included an additional table with the full set of interactions – Table A5 - in the supplementary materials, referenced in the text on page 19.
4. Thanks for the clarification of the predicted probabilities. I was wondering why PPs were not reported for the scales of “difficulty orientations”?
We did not generate predicted probabilities for the difficulty orientation scales in Tables 3 and 4 because the mimrgns and margins commands can only produce this statistic for factor or categorical variables. We appreciate this question and the thoughtful comments throughout the review process, which have enhanced the manuscript.